# Tyrosine phosphorylation of both STAT5A and STAT5B is necessary for maximal IL-2 signaling and T cell proliferation

Jian-Xin Lin [1,7], Meili Ge[1,5,7], Cheng-yu Liu[2], Ronald Holewinski[3], Thorkell Andresson[3], Zu-Xi Yu[4], Tesfay Gebregiorgis[1], Rosanne Spolski[1], Peng Li[1,6] & Warren J. Leonard [1]

Cytokine-mediated STAT5 protein activation is vital for lymphocyte development and function. In vitro tyrosine phosphorylation of a C-terminal tyrosine is critical for activation of STAT5A and STAT5B; however, the importance of STAT5 tyrosine phosphorylation in vivo has not been assessed. Here we generate Stat5a and Stat5b tyrosine-to-phenylalanine mutant knockin mice and find they have greatly reduced CD8+ T-cell numbers and profoundly diminished IL-2-induced proliferation of these cells, and this correlates with reduced induction of Myc, pRB, a range of cyclins and CDKs, and a partial G1→S phase-transition block. These mutant CD8+ T cells also exhibit decreased IL-2-mediated activation of pERK and pAKT, which we attribute in part to diminished expression of IL-2Rβ and IL-2Rγ. Our findings thus demonstrate that tyrosine phosphorylation of both STAT5A and STAT5B is essential for maximal IL-2 signaling. Moreover, our transcriptomic and proteomic analyses elucidate the molecular basis of the IL-2-induced proliferation of CD8+ T cells.

STAT5 was first identified as mammary gland factor (MGF), a transcription factor that mediates prolactin-induced β-casein expression[1]. Subsequently, it was discovered that STAT5 represents two proteins, STAT5A and STAT5B, that are more than 90% identical at the amino acid level and encoded by adjacent genes on human chromosome 17q11.2[2] and mouse chromosome 11[3]. STAT5A and STAT5B are activated by multiple cytokines, including four members of the common γ chain (γc) family of cytokines[4], IL-2, IL-7, IL-9, and IL-15[2,4,5], the IL-7-related cytokine TSLP[6], hematopoietic cytokines IL-3, IL-5 and GM-CSF[7], as well as erythropoietin, growth hormone, and prolactin[8]. Studies of Stat5a and Stat5b knockout (KO) mice and of loss-of-function mutations of STAT5B in humans underscore the importance of STAT5A in proliferation of the mammary ductal epithelium[9] and of STAT5B for

sexually dimorphic growth[10], as well as both STAT5 proteins for normal immune homeostasis and function of T and NK cells[4,11]; in addition, gain-of-function mutations of STAT5B can be associated with malignancies in humans[12].

A key mechanism in cytokine signaling is the JAK-STAT pathway, in which cytokines activate cytokine receptor-associated JAK kinases that phosphorylate a C-terminal tyrosine residue on STAT proteins[13,14]. Two phosphorylated STAT molecules then form dimers via bivalent interactions between the SH2 domain on each STAT monomer with the phosphotyrosine on the other STAT monomer. STAT dimers then translocate into the nucleus, bind to motifs in promoter and/or enhancer regions of target genes, and regulate their expression[13–15]. Analysis of the phosphoproteomic profile in IL-2-stimulated cytotoxic

[1]Laboratory of Molecular Immunology and Immunology Center, National Heart, Lung, and Blood Institute, National Institutes of Health, Bethesda, MD 20892-1674, USA. [2]Transgenic Mouse Core Facility, National Heart, Lung, and Blood Institute, National Institutes of Health, Bethesda, MD 20892-8018, USA. [3]Leidos Biomedical Research, Inc., Frederick National Laboratory for Cancer Research, Frederick, MD 21701, USA. [4]Pathology Core, National Heart Heart, Lung, and Blood Institute, National Institutes of Health, Bethesda, MD 20892, USA. [5]Present address: State Key Laboratory of Experimental Hematology, Institute of Hematology & Blood Diseases Hospital, Chinese Academy of Medical Sciences & Peking Union Medical College, Tianjin 300020, PR China. [6]Present address: Amgen, Inc., 2301 Research Blvd., Rockville, MD 20850, USA. [7]These authors contributed equally: Jian-Xin Lin, Meili Ge. ✉e-mail: linjx@nhlbi.nih.gov; leonardw@nhlbi.nih.gov

T cells showed that IL-2-induces phosphorylation of STAT5A Tyr 694 and STAT5B Tyr 699 and that this was prevented by a JAK inhibitor[16]. Interestingly, STAT5 dimers can form tetramers via N-domain interactions, with binding to tandemly-linked consensus and nonconsensus motifs, and the STAT5 tetramers have important roles in CD8[+] T cell proliferation, NK differentiation and survival, and autoimmune neuroinflammation[17–19]. Moreover, tyrosine-unphosphorylated STAT proteins, including uSTAT1, uSTAT3, uSTAT5, and uSTAT6, can directly or indirectly regulate gene expression[20,21]. uSTAT5 was reported to suppress gene expression by colocalizing with CTCF to restrain megakaryocytic differentiation[22] and suppress tumor growth by binding to heterochromatin protein 1α to stabilize heterochromatin[23]. *Stat5a*[9,24,25], *Stat5b*[10,25,26], and *Stat5a/Stat5b*[27–30] KO mouse models have revealed redundant and nonredundant roles for STAT5A and STAT5B. However, the role of tyrosine phosphorylation of STAT5 in transducing cytokine signals has not been investigated in vivo.

Here, we used CRISPR-*Cas9* gene editing to generate STAT5A and STAT5B homozygous and heterozygous tyrosine to phenylalanine mutant mouse lines to explore the role of STAT5 tyrosine phosphorylation in vitro and in vivo, using transcriptomics and proteomics as well as functional analyses. Unexpectedly, we found more severe defects in T and NK cells from *Stat5b* KI and *Stat5a* KI mice than were reported in *Stat5b* KO and *Stat5a* KO mice. Our findings indicate that tyrosine phosphorylation of both STAT5A and STAT5B is required for the full activity of IL-2, with pYSTAT5B playing a greater role. They also indicate that tyrosine-mutated STAT5A and STAT5B each can act as a dominant-negative mutant that interferes with the function of wild-type pYSTAT5 proteins. Overall, our data provide a valuable resource of IL-2-regulated genes and proteins and the impact of tyrosine phosphorylation of STAT5, thereby indicating new ways for modulating the pathways that control IL-2 signaling.

## Results

### Generation of STAT5A[Y694F] KI, STAT5B[Y699F] KI, and STAT5[Y694F/Y699F] DKI mice

Based on gene knockout studies, it is well-established that STAT5 proteins are vital for the development, differentiation, expansion, survival, and/or function of a broad range of cells in vivo[24–28,31]. In vitro studies have shown that phosphorylation of Y694 on STAT5A and Y699 on STAT5B are essential for the formation of STAT5A and STAT5B homodimers, respectively, in response to cytokine stimulation[2]. To investigate the role of tyrosine phosphorylation of STAT5 proteins in vivo, we generated homozygous mice carrying STAT5A[Y694F] KI or STAT5B[Y699F] KI alleles, as well as heterozygous mice with STAT5A[Y694F] and STAT5B[Y699F] DKI alleles (the DKI homozygous mice were not viable as discussed below) (Supplementary Fig. 1a, b show the sequences of the mutated *Stat5a* KI and *Stat5b* KI alleles generated, and Supplementary Table 1 shows the oligonucleotides used).

We first confirmed that STAT5A and STAT5B proteins were expressed at similar levels in T cells from mutant and WT littermate control mice by immunoprecipitating with antibodies to STAT5A (Fig. 1a, b, lanes 1-4) or STAT5B (lanes 5-8) and western blotting with anti-STAT5A (top panels) or anti-STAT5B (middle panels). As expected, IL-2-induced tyrosine phosphorylated STAT5A (pYSTAT5A) (Fig. 1a, bottom panel, lane 2 vs. 1) and pYSTAT5B (Fig. 1b, bottom panel, lane 6 vs. 5) were detected in WT cells but not in cells in which STAT5A Y694 or STAT5B Y699 were mutated (Fig. 1a, bottom panel, lanes 3 and 4 for *Stat5a* KI T cells and Fig. 1b, bottom panel, lanes 7 and 8 for *Stat5b* KI T cells). STAT5B was co-immunoprecipitated by anti-STAT5A in WT lysates (Fig. 1a, b, middle panels, lane 2) but not in lysates from cells not stimulated with IL-2 or from mutant cells (lanes 1, 3, and 4). Thus, tyrosine-mutated STAT5A and STAT5B are expressed at similar levels to WT STAT5 proteins, but they cannot be activated by IL-2 to form stable heterodimers, indicating that bivalent SH2-pTyr interactions are required for stable STAT5 heterodimerization.

Consistent with the phenotypes observed in homozygous *Stat5a* KO, homozygous *Stat5b* KO, and heterozygous *Stat5a/Stat5b* DKO mice[9,10,24,26–28], homozygous *Stat5a* KI, homozygous *Stat5b* KI, and heterozygous *Stat5a/Stat5b* DKI mice were viable and fertile. Unexpectedly, however, pups produced by *Stat5a/Stat5b* DKI heterozygous female mice (even when mated with WT males) did not survive. This was apparently due to starvation and dehydration (i.e., no visible milk spot and wrinkled skin), indicating that the *Stat5a/Stat5b* DKI heterozygous female mice could not properly care for/nurse their pups. To determine whether *Stat5a/Stat5b* DKI homozygous fetuses could develop in utero, we next performed timed-mating experiments. *Stat5a/Stat5b* DKI homozygous fetuses indeed developed but were pale in color and smaller in size than the WT and heterozygous fetuses (Supplementary Fig. 1c), indicating severe anemia and possibly other defects as well, consistent with the phenotype of *Stat5* DKO mice[27,28]. *Stat5a* KI female mice also exhibited severe defects in the expansion of the mammary ductal epithelium during late pregnancy (Supplementary Fig. 1d), consistent with the phenotype of *Stat5a* KO females[9]. Moreover, 3–4 month old as well as 1–2 month old *Stat5b* KI male mice had significantly lower body weight compared to WT controls (Supplementary Fig. 1e), whereas the 3–4 month old females had weights more similar to WT controls (Supplementary Fig. 1f), consistent with the sexually dimorphic growth found in *Stat5b* KO mice[10], whereas such changes were not observed in *Stat5a* KI mice (Supplementary Fig. 1g, h). Thus, the tyrosine-mutated *Stat5a* KI and *Stat5b* KI mice exhibit phenotypic similarities to those of the corresponding *Stat5a* KO and *Stat5b* KO mice.

### Fewer T and NK cells in tyrosine mutant *Stat5a* KI and *Stat5b* KI mice

Because STAT5 proteins play vital roles in mediating the actions of a number of $\gamma_c$-family cytokines that collectively are important for the development, differentiation, expansion, survival, and function of T, B, and NK cells[32,33], we examined the immune phenotype in the STAT5 tyrosine mutant KI mice. Thymocyte numbers were lower in *Stat5a* KI and *Stat5b* KI homozygous mice but similar in *Stat5a/Stat5b* DKI heterozygous (het) mice to their respective WT littermate controls (Fig. 1c and Supplementary Fig. 2a). The frequencies of CD4[-]CD8[-] double negative (DN), CD4[+]CD8[+] double positive (DP), and single positive CD4[+] and CD8[+] thymocytes in *Stat5a* KI and *Stat5b* KI homozygous mice and *Stat5a/Stat5b* DKI het mice were similar to WT littermates (Supplementary Fig. 2b, c, first set of three bar graphs), but CD8[+] thymocyte numbers were lower in *Stat5a* KI and *Stat5b* KI homozygous mice, resulting in decreased numbers of some of the populations (e.g., DN, DP, and CD8[+] T cells) in *Stat5a* KI and *Stat5b* KI homozygous mice but not in *Stat5a/Stat5b* DKI het mice (Fig. 1c and Supplementary Fig. 2c, second set of three bar graphs).

Total splenic cell numbers were lower in all three KI lines, particularly in the *Stat5b* KI mice (Fig. 1c and Supplementary Fig. 2d). Thus, although B220[+] and CD3[+] cell frequencies were not significantly altered, the numbers of these cells were lower in all three KI lines, and T and B cell numbers were lowest in *Stat5b* KI homozygous mice (Fig. 1c and Supplementary Fig. 2e, f). CD4[+] T cell numbers were not lower in *Stat5a* KI but were significantly lower in *Stat5b* KI mice (Figs. 1c and 2b) but not *Stat5a/Stat5b* DKI het mice (Fig. 1c and Supplementary Fig. 2g, h), whereas CD8[+] T cells were decreased in all three lines, particularly in *Stat5b* KI mice (Figs. 1c and 2a, b, Supplementary Fig. 2g, h). In addition, CD4[+]CD25[+] cell numbers were diminished in all 3 lines and CD4[+]FoxP3[+] Tregs were decreased in *Stat5b* KI mice and trended lower in *Stat5a* KI mice but not in *Stat5a/Stat5b* DKI het mice (Figs. 1c, 2c–f, and Supplementary Fig. 2i, j). The frequencies and numbers of CD8[+]CD122[hi] T cells were lower in all three mutant mouse lines (Figs. 1c and 2g, h, and Supplementary Fig. 2k, l). Thus, tyrosine phosphorylation of both STAT5A and STAT5B is required for normal peripheral T cell homeostasis. Given

the relatively normal T cell numbers in both *Stat5a* KO or *Stat5b* KO mice[30], the more severe defects observed in *Stat5a* KI or *Stat5b* KI mice indicate that both STAT5A and STAT5B tyrosine mutants interfered with WT STAT5, suggesting a dominant negative effect.

We next examined whether there were also defects in NK cells in the KI mice. Indeed, consistent with the phenotype of *Stat5* KO mice[29], total bone marrow cells (Fig. 1c and Supplementary Fig. 3a) were significantly decreased in *Stat5b* KI mice; both bone marrow total NK cells (lin⁻CD122⁺) (Fig. 1c and Supplementary Fig. 3b, c) and mature (lin⁻CD122⁺NK1.1⁺DX5⁺) NK cell numbers (Fig. 1c and Supplementary Fig. 3d, e) were significantly decreased in both *Stat5a* KI and *Stat5b* KI mice. Splenic NK cells (CD3⁻CD122⁺NK1.1⁺) were significantly decreased in all three tyrosine mutant KI mouse lines (Fig. 1c and Supplementary Fig. 3f, g). To assure that we were accurately identifying spleen NK cells, we compared CD122 with two additional NK cell markers (DX5 and NKp46) and found that all three NK markers similarly stained NK cells from WT and *Stat5a* KI and *Stat5b* KI mice (Supplementary Fig. 3h, i). Thus, both tyrosine phosphorylated STAT5A and STAT5B are required for normal numbers of NK cells as well as CD8⁺ T cells.

To determine if the mutant T cells and especially mutant CD8⁺ T cells, were indeed defective in response to IL-2-induced proliferation in vivo, PBS or MSA-conjugated IL-2 was intraperitoneally injected into *Stat5a* KI, *Stat5b* KI, and their corresponding WT littermate mice on days 0, 3, and 6. Splenocytes were isolated on day 7 and T cells were examined by flow cytometric analysis. Consistent with significantly decreased cellularity of T cells in *Stat5a* KI and *Stat5b* KI mice, WT CD3⁺ T cells and CD3⁺CD8⁺ T cells had more SSC^high cells than observed for corresponding cells from the *Stat5* KI mice or from mice injected with PBS (Fig. 1k, l, dot plots), indicating less expansion of the *Stat5* KI cells. Correspondingly, there were significantly fewer T cells and especially CD8⁺ T cells in the IL-2-injected *Stat5* KI mice than in the IL-2-injected WT mice (Fig. 2i, j, bar graphs). These data indicate that the decreased T cell and CD8⁺ T cell numbers in mutant mice are, at least in part, due to the defective IL-2-induced proliferation.

## Defective IL-2-induced proliferation and cell cycle progression of *Stat5a* and *Stat5b* KI CD8⁺ T cells

Given that IL-2 drives the expansion of T cells[34,35] and that CD8⁺ and CD8⁺CD122^hi T cell numbers were decreased in all three KI mouse lines

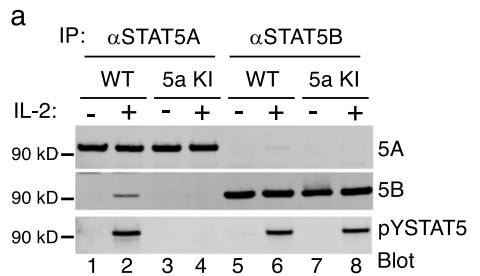

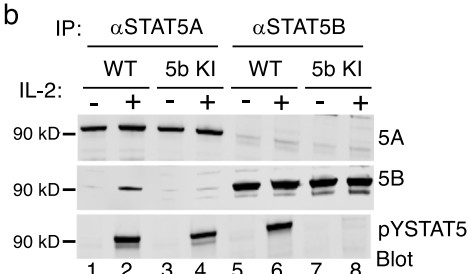

| | | Immune cellularity in mutant mice (p-value) | | |
|---|---|---|---|---|
| | | *Stat5a* KI | *Stat5b* KI | *Stat5a/b* het |
| Total cell numbers | Thymus | ↓↓ 0.0041 | ↓↓ 0.0033 | ↔ 09932 |
| | Bone marrow | ↔ 0.2462 | ↓ 0.0173 | ↔ 0.3018 |
| | Spleen | ↓↓ 0.0029 | ↓↓↓↓ <0.0001 | ↓ 0.0120 |
| Thymus | DN | ↓↓ 0.0024 | ↓↓ 0.0095 | ↔ 0.2300 |
| | CD4⁺CD8⁺ | ↓↓ 0.0045 | ↓↓ 0.0077 | ↔ 0.5484 |
| | CD4⁺CD8⁻ | ↔ 0.0593 | ↓↓ 0.0038 | ↔ 0.3162 |
| | CD4⁻CD8⁺ | ↓↓ 0.0027 | ↓↓ 0.0012 | ↔ 0.9345 |
| Bone marrow | Total NK | ↓ 0.0450 | ↓↓ 0.0034 | ↔ 0.1344 |
| | Mature NK | ↓ 0.0488 | ↓↓ 0.0017 | ↔ 0.1745 |
| Spleen | B cells | ↓↓ 0.0042 | ↓↓↓↓ <0.0001 | ↓ 0.0256 |
| | T cells | ↓ 0.0351 | ↓↓↓↓ <0.0001 | ↓↓ 0.0065 |
| | CD4⁺ | ↔ 0.2390 | ↓↓↓↓ 0.0002 | ↔ 0.0633 |
| | CD8⁺ | ↓↓↓ 0.0002 | ↓↓↓↓ <0.0001 | ↓↓↓↓ 0.0001 |
| | CD4⁺CD25⁺ | ↓↓↓ 0.0003 | ↓↓↓↓ <0.0001 | ↓ 0.0410 |
| | CD8⁺CD122^hi | ↓↓↓↓ <0.0001 | ↓↓↓↓ <0.0001 | ↓↓↓ 0.0003 |
| | CD4⁺Foxp3⁺ | ↓ 0.0238 | ↓↓ 0.0053 | ↔ 0.7637 |
| | Total NK | ↓↓↓↓ <0.0001 | ↓↓↓↓ <0.0001 | ↓↓↓↓ <0.0001 |

**Fig. 1 | Normal STAT5 protein expression but fewer T, B, and NK cells in *Stat5a* KI and especially in *Stat5b* KI mice. a, b** Immunoblots of anti-STAT5A, anti-STAT5B, and anti-pSTAT5 antibodies. **a** Immunoprecipitation of total cell lysates from control or IL-2-stimulated WT and *Stat5a* KI T cells with anti-STAT5A and anti-STAT5B, respectively, and blotting with antibodies to STAT5A, STAT5B, and pYSTAT5. **b** Immunoprecipitation of total cell lysates from control or IL-2 stimulated WT and *Stat5b* KI T cells followed by blotting with anti-STAT5A, anti-STAT5B, and anti-pYSTAT5. Source data for **a** and **b** are provided as a Source Data File.

**c** Summary of cellularity of *Stat5a* KI, *Stat5b* KI, and *Stat5a/b* heterozygous thymocytes, bone marrow NK cells, and splenic T, B, and NK cells. The numbers are the *p*-values that are derived from Fig. 2b, d, f, h, j, Supplementary Fig. 2a, c, d, f, h, j, l, and Supplementary Fig. 3a, c, e, g (In the source data file, the raw data are shown separately for each of these main and Supplementary Figs. rather than for **c**); the red arrows show statistically significant decreases in cells in KI as compared to WT mice, and the horizontal black arrow lines indicate populations that are not significantly different in KI and WT. Source data are provided as a Source Data File.

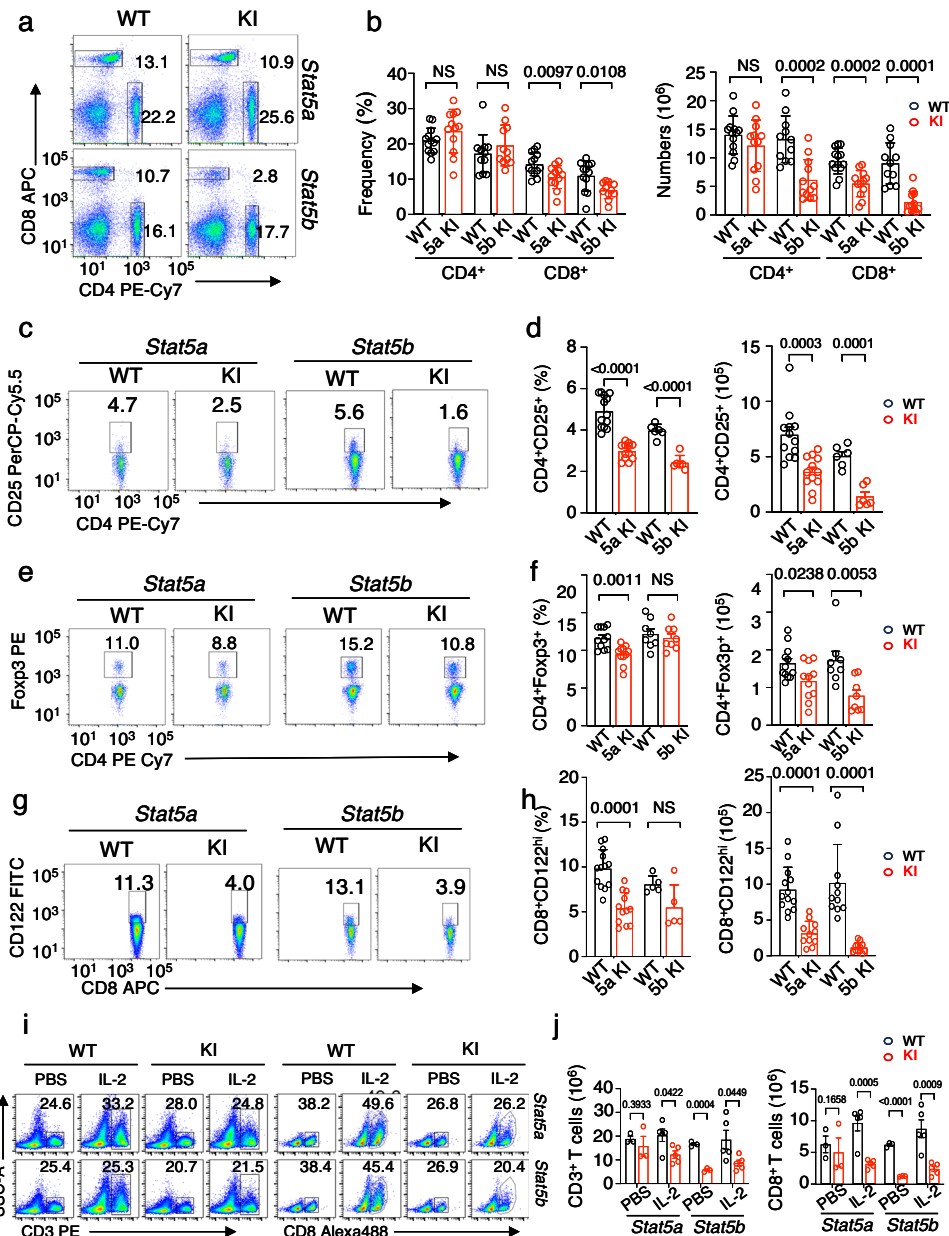

**Fig. 2 | Fewer CD8⁺ T cells, and decreased CD4⁺CD25⁺ and CD8⁺CD122ʰⁱ T cell numbers in *Stat5a* KI and *Stat5b* KI mice.** **a** Representative flow cytometric profiles of splenic CD4⁺ and CD8⁺ T cells in WT and *Stat5a* KI (top panels) and WT and *Stat5b* KI (bottom panels) mice. **b** Summary of the frequencies (left graph) and numbers (right graph) of splenic CD4⁺ and CD8⁺ T cells in WT and *Stat5a* KI and *Stat5b* KI mice (*n* = 13 for *Stat5a* WT CD4⁺ T cells, *n* = 12 for *Stat5a* KI CD4⁺ T cells, *n* = 12 for *Stat5b* WT CD4⁺ T cells, *n* = 12 for *Stat5b* KI CD4⁺ T cells, *n* = 13 for *Stat5a* WT CD8⁺ T cells, *n* = 12 for *Stat5a* KI CD8⁺ T cells, *n* = 11 for *Stat5b* WT CD8⁺ T cells, and *n* = 12 for *Stat5b* KI CD8⁺ T cells). **c** Representative flow cytometric profiles of splenic CD4⁺CD25⁺ T cells in WT and *Stat5a* KI (left panels) and WT and *Stat5b* KI (right panels) mice. **d** Summary of the frequencies (left graph) and numbers (right graph) of spleen CD4⁺CD25⁺ T cells in WT and *Stat5a* KI and *Stat5b* KI mice (*n* = 12 for both *Stat5a* WT and *Stat5a* KI mice and *n* = 6 for both *Stat5b* WT and *Stat5b* KI mice). **e** Representative flow cytometric profiles of spleen CD4⁺Foxp3⁺ T cells in WT and *Stat5a* KI (left panels) and WT and *Stat5b* KI (right panels) mice. **f** Summary of the frequencies (left graph) and numbers (right graph) of spleen CD4⁺Foxp3⁺ T cells in WT and *Stat5a* KI and *Stat5b* KI mice (*n* = 12 for *Stat5a* WT mice, *n* = 11 for *Stat5a*

KI mice, *n* = 8 for both *Stat5b* WT and *Stat5b* KI mice). **g** Representative flow cytometric profiles of spleen CD8⁺CD122ʰⁱ T cells in WT and *Stat5a* KI (left panels) and WT and *Stat5b* KI (right panels) mice. **h** Summary of the frequencies (left graph) and numbers (right graph) of spleen CD8⁺CD122ʰⁱ T cells in WT and *Stat5a* KI and *Stat5b* KI mice (*n* = 13 for *Stat5a* WT, *n* = 11 for *Stat5a* KI, *n* = 11 for *Stat5b* WT, and *n* = 12 for *Stat5b* KI). **i** Representative flow cytometric profiles of T (left four panels) and CD8⁺ T cells (right four panels) in WT and *Stat5a* KI (upper panels) and *Stat5b* KI (lower panels) mice on day 7 after PBS or IL-2 injection. **j** Bar graphs showing numbers of CD3⁺ and CD3⁺CD8⁺ T cells in WT and *Stat5a* KI and *Stat5b* KI mice on day 7 after PBS (*n* = 3) or IL-2 injection (*n* = 5). The gating strategies for **a, c, e, g, and i** are shown in Supplementary Fig. 8. The numbers in **a, c, e, g, and i** are percentage of gated cells; in graphs **b, d, f, h, and j**, black and red open bars represent WT and mutants, respectively, and the error bars (SD) and *p*-values are shown, which were determined by multiple unpaired *t* test using two-stage step-up method of Benjamini, Krieger and Yekutieli. Source data for **b, d, f, h, and j** are provided as a Source Data File.

(Fig. 1c), we next sought to elucidate the underlying mechanisms for the decreased CD8⁺ T cell numbers in the mutant mice. We first assessed the IL-2 responsiveness of freshly isolated CD8⁺ T cells. STAT5 was activated by $10^{-10}$ M IL-2 in freshly isolated WT CD8⁺ T cells, with

maximal responsivness at $10^{-8}$ M (~500 IU/ml) IL-2 (Supplementary Fig. 4a, b), a concentration approximately 10-fold higher than the $K_d$ of intermediate-affinity IL-2 receptors. We decided to use 500 IU/ml IL-2 in all subsequent experiments. *Stat5a* KI and *Stat5b* KI CD8⁺ T cells had

markedly decreased IL-2-induced proliferation based on CFSE (carboxyfluorescein succinimidyl ester) dilution (Fig. 3a, left panels) but had similar CFSE dilution to that of WT control cells in response to TCR (anti-CD3 + anti-CD28) (Fig. 3a, right panels). We next evaluated the stage(s) at which IL-2-induced cell cycle progression was affected by using Click-iT EdU (5-Ethynyl-2-deoxyuridine) incorporation assays (schematic in Fig. 3b). IL-2 stimulation of WT CD8$^+$ T cells over a 3-day period resulted in a transition from G0/G1 phase cells to increased S phase cells (Fig. 3c, d black lines), whereas more G0/G1 phase cells and fewer S phase cells were detected in *Stat5a* KI and especially *Stat5b* KI CD8$^+$ T cells (Fig. 3c, d, red lines), indicating defective IL-2-induced cell cycle progression.

In quiescent cells, hypo-phosphorylated RB is known to bind to E2F-DP1, which suppresses the transcription of E2F target genes, and upon mitogen stimulation, RB is progressively phosphorylated (pRB) by CDK4/CDK6-CCND and CDK2-CCNE, leading to the disassociation of pRB from E2F-DP1 and transcription of E2F-target genes (schematic in Fig. 3e), including cell cycle progression-related genes, which allows cell cycle entry and the initiation of DNA synthesis[36,37]. In WT CD8$^+$ T cells, increased pRB was detected after one day of IL-2 stimulation and further increased over 3 days (Fig. 3f, g, left panels; Fig. 3h, i black lines). In contrast, fewer pRB$^+$ cells and lower pRB levels were detected in *Stat5a* KI and *Stat5b* KI CD8$^+$ T cells (Fig. 3f, g, right panels; Fig. 3h, i red lines), revealing a role for IL-2-induced pYSTAT5 in augmenting RB phosphorylation and driving cell cycle progression.

### Lower ERK and AKT activation in *Stat5a* KI and *Stat5b* KI CD8$^+$ T cells and decreased virtual memory splenic CD8$^+$ T cells in *Stat5a* and *Stat5b* KI mice

Given the defective IL-2-induced proliferation in *Stat5a* KI and *Stat5b* KI CD8$^+$ T cells, we analyzed these cells for defects in activation not only of STAT5 but also of ERK and AKT, which collectively mediate T cell proliferation and survival[34,35,38]. In WT CD8$^+$ T cells, IL-2 rapidly activated pSTAT5, with sustained pSTAT5 for at least 22 hr; pERK and pAKT also were rapidly and strongly increased (black lines in Fig. 4a–c for pYSTAT5, pERK, and pAKT, respectively; Supplementary Fig. 4c–e show representative flow plots). As expected, both pYSTAT5A and pYSTAT5B contributed to the pYSTAT5 signal, with lower IL-2-induced pYSTAT5 levels in both *Stat5a* and *Stat5b* KI cells than in the corresponding WT cells (Fig. 4a, red lines, Supplementary Fig. 4c). Interestingly, pERK and pAKT were also lower in *Stat5a* and *Stat5b* KI CD8$^+$ T cells than in WT cells, especially at 22 hr (Fig. 4b, c, red lines and Supplementary 4d, e), suggesting that pYSTAT5 is involved in the activation or sustaining the activation of these pathways.

Intermediate and high affinity IL-2 receptors transduce IL-2 signals via heterodimerization of the IL-2Rβ and IL-2Rγ cytoplasmic domains[39,40], thereby activating JAK1/JAK3-STAT5, ERK, and PI 3-kinase/AKT pathways[34,35,38]. In naïve mice, CD44$^{hi}$CD122$^{hi}$ memory CD8$^+$ T cells are the major population proliferating to high dose IL-2[41]. Most of these cells are CD8$^+$CD44$^{hi}$CD122$^{hi}$CD49d$^{lo}$ antigen-naïve virtual memory cells, which require γ$_c$ family cytokines, including IL-4, IL-7, and IL-15 for their development and maintenance[42–44]. The lower pERK and pAKT levels induced by IL-2 in *Stat5a* KI and *Stat5b* KI CD8$^+$ T cells prompted us to evaluate whether decreased numbers of memory and/or virtual memory CD8$^+$ T cells might explain the decreased proliferation. Indeed, CD8$^+$CD44$^{hi}$ memory cell numbers in *Stat5a* KI or *Stat5b* KI mice were reduced (Fig. 4d, e), as were CD8$^+$CD122$^{hi}$ cells (Fig. 4f, g) and CD8$^+$CD44$^{hi}$CD122$^{hi}$CD49d$^{lo}$ virtual memory cells (Fig. 4h, i). CD8$^+$CD44$^{hi}$ memory phenotype cells can be further defined by CD122$^{lo}$ and CD122$^{hi}$ populations. MHC I is required for the expansion of CD8$^+$CD44$^{hi}$CD122$^{lo}$ cells whereas γ$_c$-family cytokines, including IL-2, are required for the expansion of CD8$^+$CD44$^+$CD122$^{hi}$ cells[45,46]. In mutant mice, there was a significant increase of CD8$^+$CD44$^{hi}$CD122$^{lo}$ cells and decrease of CD8$^+$CD44$^{hi}$CD122$^{hi}$ cells (Fig. 4j, k), which is consistent with normal TCR-induced proliferation but markedly defective IL-2-induced proliferation of freshly isolated *Stat5a* KI and *Stat5b* KI CD8$^+$ T cells (Fig. 3a). Thus both pYSTAT5A and pYSTAT5B are required for normal numbers of memory and virtual memory CD8$^+$ T cells.

### IL-2-induced expression of a broad range of genes requires pYSTAT5

Having demonstrated severe defects in proliferation, cell cycle progression, and pERK and pAKT activation in tyrosine mutant STAT5 CD8$^+$ T cells, we next sought to elucidate the molecular basis for the defects observed in the STAT5 tyrosine mutant CD8$^+$ T cells. Using RNA-seq analysis, we compared differentially expressed genes (DEGs) in freshly isolated WT, *Stat5a* KI, and *Stat5b* KI CD8$^+$ T cells in response to IL-2 stimulation for 4, 24, and 48 hrs. A principal component analysis (Supplementary Fig. 5a) showed that mRNAs expressed in control samples from WT and mutant cells were clustered together, whereas there was variation in mRNA expression in WT and mutant cells stimulated by IL-2 that was greatest at 48 hr. Using criteria of FC ≥ 2 and FDR ≤ 0.05, a total of 3639 IL-2-regulated mRNAs in WT cells were identified (Supplementary Data 1). When we compared each KI to its respective WT, we found for both the *Stat5b* KI and *Stat5a* KI cells that more genes were highly expressed in WT than in the KI cells, with a total of 741 mRNAs having lower expression (FC ≥ 2) in *Stat5a* KI and/or *Stat5b* KI cells than in WT CD8$^+$ T cells (Supplementary Data 2). Consistent with a greater defect in *Stat5b* KI cells, there were more DEGs in the *Stat5b* KI cells, and almost all mRNAs with lower expression in *Stat5a* KI CD8$^+$ T cells overlapped with those with lower expression in *Stat5b* KI CD8$^+$ T cells (Fig. 5a). Strikingly, a Gene Set Enrichment Analysis (GSEA) revealed that the top 10 hallmark genesets dysregulated in both *Stat5a* KI and *Stat5b* KI cells were related to cell cycle progression (Fig. 5b), including hallmark genesets for MYC, E2F, G2M, and MTORC1 (Fig. 5c). These included DEGs encoding transcription factors, factors for rate-limiting enzymes for deoxynucleotide synthesis from ribonucleotides[47,48], DNA replication[49], DNA repair[50,51], transcription initiation and elongation[52], translation initiation and elongation[53–55], and solute carriers[56] (Fig. 5d and Supplementary Data 2), as well as genes encoding for proteosome components[57–59], nuclear import and export[60] (Supplementary Fig. 5b and Supplementary Data 2), and mitochondrial components[61] (Supplementary Data 2). Thus, pYSTAT5 is critical for transducing the IL-2 mitogenic signal by regulating expression of a range of genes encoding essential components of cell cycle progression and proliferation.

Interestingly, only two IL-2-induced transcription factors, *Crem* and *Myc*, were rapidly induced at 4 hr of IL-2 stimulation; *Crem* was only detected at the 4 hr time point, whereas *Myc* was expressed at high levels for up to 48 hrs (Fig. 5d, highlighted in red with asterisks), consistent with our prior observation[18]. mRNAs for other transcription factors, including *E2f1*, *E2f2*, *E2f3*, and *Tfdp1* were detected at 24 hr and were higher at 48 hr (Fig. 5d, in red), consistent with their being downstream of Myc[62]. Together with the rapid and sustained induction by IL-2 of *Myc* in WT cells but lower induction of *Myc* in *Stat5* KI cells, these data indicate the important role(s) of Myc in relaying and amplifying the IL-2-pYSTAT5 signal to downstream targets. In addition, the expression levels of known IL-2-inducible mRNAs were significantly decreased in mutant cells, including those encoding cytokines, cytokine receptors, and effector molecules (Supplementary Data 2, e.g., *Ifng*, *Tnf*, *Lta*, *Lif*, *Osm*, *Il16*, *Il2ra*, *Il2rb*, *Il12b2*, and *Gzmb*)[18,63]. Interestingly, *Il2rg* was induced by IL-2 albeit more weakly than *Il2ra* and *Il2rb*, and its induction was not dependent on pYSTAT5 (Supplementary Fig. 5c).

To identify pYSTAT5-target genes, we next performed ChIP-seq analysis using freshly isolated CD8$^+$ T cells from the KI and corresponding WT mice that were not stimulated or stimulated with IL-2 for 4 hrs. We used a pan-STAT5 antibody that recognizes both STAT5A and STAT5B, which allowed us to estimate STAT5A binding sites in *Stat5b*

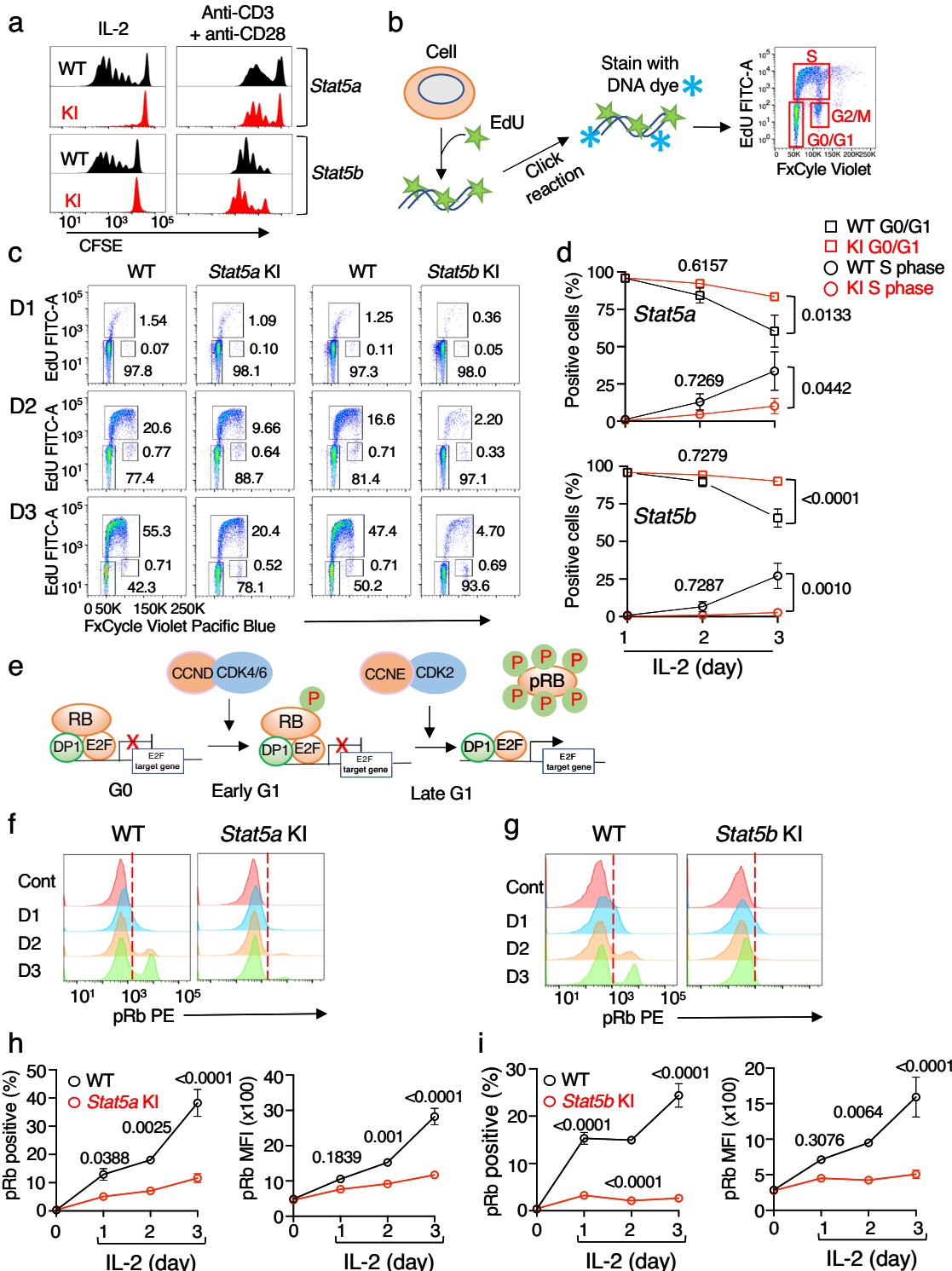

KI cells and STAT5B binding sites in *Stat5a* KI cells. Of the top 500 STAT5 binding peaks we identified in WT or *Stat5a* KI or *Stat5b* KI CD8[+] T cells, more than >90% contained GAS motifs (Supplementary Fig. 5d). There were a total of 9564 STAT5 binding sites in the *Stat5a* WT and 9199 STAT5 binding sites in the *Stat5b* WT CD8[+] T cells (i.e., identified in the WT mice corresponding to each of the mutant mouse lines; see Supplementary Fig. 5e), with 7513 peaks shared by both. Of these, only 353 (4.7%) peaks were significantly altered in *Stat5a* KI CD8[+] T cells, in contrast to the greater loss of STAT5 binding peaks (3546, 47.1%) in *Stat5b* KI CD8[+] T cells (Supplementary Fig. 5f). A greater genome-wide decrease in STAT5 binding was observed in

*Stat5b* KI than in *Stat5a* KI CD8[+] T cells (Fig. 5e, f). Given that the tyrosine mutated STAT5A and STAT5B proteins cannot form homo-dimers and bind to GAS motifs[2], the residual STAT5 binding in *Stat5a* KI cells likely represented chromatin-bound pYSTAT5B homodimers and that in *Stat5b* KI cells likely represented pYSTAT5A homodimers. Consistent with the more severe defects observed in *Stat5b* KI than in *Stat5a* KI CD8[+] T cells, the greatly reduced STAT5 binding peaks in the *Stat5b* KI cells suggested that STAT5B is the major STAT5 protein bound to STAT5-target genes and that it plays a greater role in trans-ducing IL-2 signals. Consistent with this, using criteria of FC ≥ 2 for both mRNA levels from RNA-seq data and for STAT5 binding from

**Fig. 3 | Defective IL-2-induced proliferation, reduced cell cycle progression, and lower levels of pRB in *Stat5a* KI and *Stat5b* KI CD8+ T cells. a** Representative flow cytometric profiles of two CFSE dilution experiments using freshly isolated WT (top and third panels), *Stat5a* KI (second panel), and *Stat5b* KI (bottom panel) CD8+ T cells were cultured with 500 IU/ml IL-2 for 4 days (left panels) or with plate bound anti-CD3 and soluble anti-CD28 for 3 days (right panels). **b** Schematic showing the Click-IT EdU assay and flow cytometric profiles of cells at the G0/G1, S, and G2/M phases of the cell cycle. **c** Representative flow cytometric profiles of two experiments show cell cycle progression of WT, *Stat5a* KI, and *Stat5b* KI splenic CD8+ T cells stimulated by 500 IU/ml of IL-2 using Click-iT EdU assays; the numbers are the percentages of gated populations. **d** Percentage of WT (black lines, n = 4), *Stat5a* KI (red lines in top graph, n = 4), and *Stat5b* KI (red lines in bottom graph, n = 3) cells at G0/G1 phase (EdU-FxCycle violet-), S phases (EdU+), and G2/M phases (EdU-FxCycle violet+). The error bars and *p*-values are shown, which were determined by Two-way ANOVA multiple comparisons using the Šídák hypothesis test. In **d**, each data point represents 4 biological replicates combined from two

independent experiments, and each biological replicate was derived from the combined cells from two *Stat5a* KI mice or three *Stat5b* KI mice. **e** Schematic showing phosphorylation of RB by CCND/CDK4/6 at early G1 phase and then by CCNE/CDK2 at late G1 phase to release pRB from DP1-E2F and turn on DP1-E2F-mediated transcription of their target genes. **f** and **g** Representative flow cytometric profiles of pRB levels from two independent experiments in WT (left panels), *Stat5a* KI (**f**, right panel) and *Stat5b* KI (**g**, right panel) CD8+ T cells during a 3-day IL-2 stimulation. The vertical dashed lines indicate the gates for the percentages of cells in **f** and **g**. The gating strategies for **a**, **c**, **f**, and **g** are shown in Supplementary Fig. 9. **h** and **i** Data were combined from two independent experiments to show the percentage of pRB+ cells and pRB levels (MFI) in CD8+ T cells from WT littermate (black lines, n = 4 for *Stat5a* and *Stat5b*), *Stat5a* KI (**h**, red line, n = 4), and *Stat5b* KI (**i**, red line, n = 4) mice; error bars (SEM) and *p*-values are shown, which were determined by multiple unpaired *t* test using the Holm-Šídák method. Source data for **d**, **h**, and **i** are provided as a Source Data File.

ChIP-seq data in WT versus mutant cells, 18 versus 123 genes were identified in *Stat5a* KI versus *Stat5b* KI CD8+ T cells, respectively, as compared to their corresponding WT cells, as direct STAT5-target genes (i.e., genes to which STAT5 bound and whose expression was diminished in *Stat5a* KI and/or *Stat5b* KI CD8+ T cells; Fig. 5g). These included well-known STAT5-target genes (e.g., *Il2ra, Il2rb, Il12rb2, Il6ra, Il6st, Cdk6, Cish, Socs2, Lta, Gzmb*, and *Lif*[18], as well as less well appreciated IL-2-induced STAT5-target genes that are involved in cell-cycle progression, including *Mov10, Setbp1, Nek6, Melk, Spdl1, Tars, Slc30a4, Scd1, Scd2, Spata5*, and *Larp1* (Supplementary Data 3). Interestingly, STAT5 binding peaks were also identified upstream of the *Myc* locus, with IL-2-induced binding in WT cells but decreased binding in *Stat5b* KI CD8+ T cells. Two canonical STAT5 binding sites were indeed found at positions −1992 and −1069 relative to the Myc transcription start site (Supplementary Fig. 5g), suggesting that STAT5B may also be involved in regulating *Myc* expression.

## Markedly diminished IL-2-induced protein levels of Myc, E2F, CCNs, and CDKs in *Stat5a* and *Stat5b* KI cells

It is well-known that Myc critically controls cell cycle progression in response to mitogen stimulation[64]. The genes encoding transcription factors E2F1, E2F2, and E2F3 are Myc target genes, and these E2F family proteins form heterodimers with dimerization partner 1 (DP1, encoded by *Tfdp1*) to regulate the expression of a large number of genes that are critical for cell cycle entry and progression[65–67], E2F proteins also regulate *Myc* expression[68] (Fig. 6a). Our observation that there was greatly reduced expression of *Myc* and of cell cycle progression genes (Fig. 5d) prompted us to examine whether their protein expression levels were also diminished in *Stat5a* KI and *Stat5b* KI CD8+ T cells in response to IL-2 stimulation. Previously, IL-2 was shown to mediate the induction of Myc in TCR-preactivated CD4+ T cells[69–71]. Here we demonstrate that Myc protein was induced by IL-2 in freshly isolated WT CD8+ T cells by 4 hr and sustained for at least 48 hr, whereas this was not observed in *Stat5a* KI and *Stat5b* KI CD8+ T cells (Fig. 6b, c, lanes 5-8 vs 1-4; quantified in the line graphs) as assessed by western blot analysis, indicating that the lower Myc expression in *Stat5* KI CD8+ T cells dampens expression of downstream genes essential for IL-2-induced proliferation of these cells.

We next evaluated the protein levels of E2F1, CCND1, CCNA2, CCNE, CDK7, CDK2, CDK4, and CDK6 in WT and mutant cells, which are critical for the initiation and control of cell cycle progression from G1→S→G2→M[36,37,72]. Consistent with the RNA-seq data (Figs. 5d and 6d), in CD8+ T cells from WT mice, the percentage of cells expressing these factors increased by day 1 of IL-2 stimulation and progressively increased at days 2 and 3 (Fig. 6e, f, black bars; representative experiments are in Supplementary Fig. 6a–h, with *Stat5a* data in upper panels and *Stat5b* data in lower panels). In contrast, induction was much lower in *Stat5a* KI CD8+ T cells and even more defective in *Stat5b* KI CD8+

T cells (Fig. 6e, f, red bars and Supplementary Fig. 6a–h). Along with cyclin H (CCNH) and MAT1, CDK7 is a key component of CDK activating kinase (CAK, Fig. 6g, left schematic)[73], which activates CDK1, CDK2, CDK4, and CDK6[74], and CDK7 also associates with TFIIH to phosphorylate proteins involved in RNA polymerase II-mediated transcription (Fig. 6g, right schematic)[75]. Interestingly, *Ccnd1* mRNA expression was suppressed by IL-2 in both WT and mutant cells, whereas *Cdk7* mRNA levels were induced by IL-2 in WT and mutant cells (Fig. 6d). Nevertheless, their protein levels were significantly induced by IL-2 in WT cells but to a lower degree in *Stat5a* KI cells and not induced in *Stat5b* KI cells (Fig. 6e, f, and Supplementary Fig. 6b, e), suggesting a posttranscriptional component to their regulation by IL-2.

## Dysregulated expression of proteins in *Stat5* KI cells by global proteomic analysis

We next investigated global changes in protein expression levels in WT and *Stat5a* KI CD8+ T cells stimulated with IL-2 over a two-day period using mass spectrometry. The peptides corresponding to a total of 6450 proteins were identified using criteria of 1% FDR (proteins in ≥ 8 samples). Using an absolute logFC ≥ 0.4 and *p*-value ≤ 0.05, we identified 524 proteins that were differentially expressed between WT and *Stat5a* KI cells. A comparison of proteins induced by IL-2 at days 1 and 2 in WT and *Stat5a* KI CD8+ T cells revealed 486 proteins whose expression was significantly lower in the KI cells (169 at day 1 and 419 at day 2, with 102 significantly lower on both days) (Supplementary Fig. 7a, Supplementary Data 4). Analysis of the dysregulated proteins showed that the top 10 enriched genesets (Supplementary Fig. 7b) were similar to those identified in RNA-seq data (Fig. 5b), including E2F-targets, MYC-targets, G2M checkpoint, and MTORC1 signaling (Supplementary Fig. 7b and Supplementary Fig. 7c for E2F-TARGETS, Supplementary Fig. 7d for MYC-TARGETS v1, Supplementary 7e for G2M_checkpoint, and Supplementary Fig. 7f for MTORC1_signaling). Consistent with our RNA-seq data, many proteins, including EOMES, CDK1, CDK2, CDK4, CDK6, AURKA, AURKB, MCM2-7, were more greatly induced by IL-2 in WT cells than in mutant cells (Fig. 7a and Supplementary Data 4). Surprisingly, a number of mRNAs were not lower or only modestly lower (FC < 2) in their expression in *Stat5a* KI cells as compared to WT cells, but their protein levels were nevertheless significantly lower in the KI than in WT cells (Fig. 7b and Supplementary Data 4), including RB family protein RBL1 (p107)[66], mediator (MED21)[76], ATP-dependent DNA helicase (DDX11)[77], and ATP-dependent RNA helicases (DDX24)[78]. Interestingly, *Il2rg* mRNA was induced by IL-2 in both WT and *Stat5a* KI CD8+ T cells, but IL2RG (also known as IL-2Rγ or γc) protein was significantly induced by IL-2 in WT cells but not in *Stat5a* KI mutant cells (Fig. 7b and Supplementary Data 4). The decreased expression of IL-2Rβ and γc (Supplementary Fig. 5c) likely contributes to the decreased gene induction/signaling in the *Stat5a* KI CD8+ T cells. Collectively, these data provide insights into the molecular mechanisms controlling the

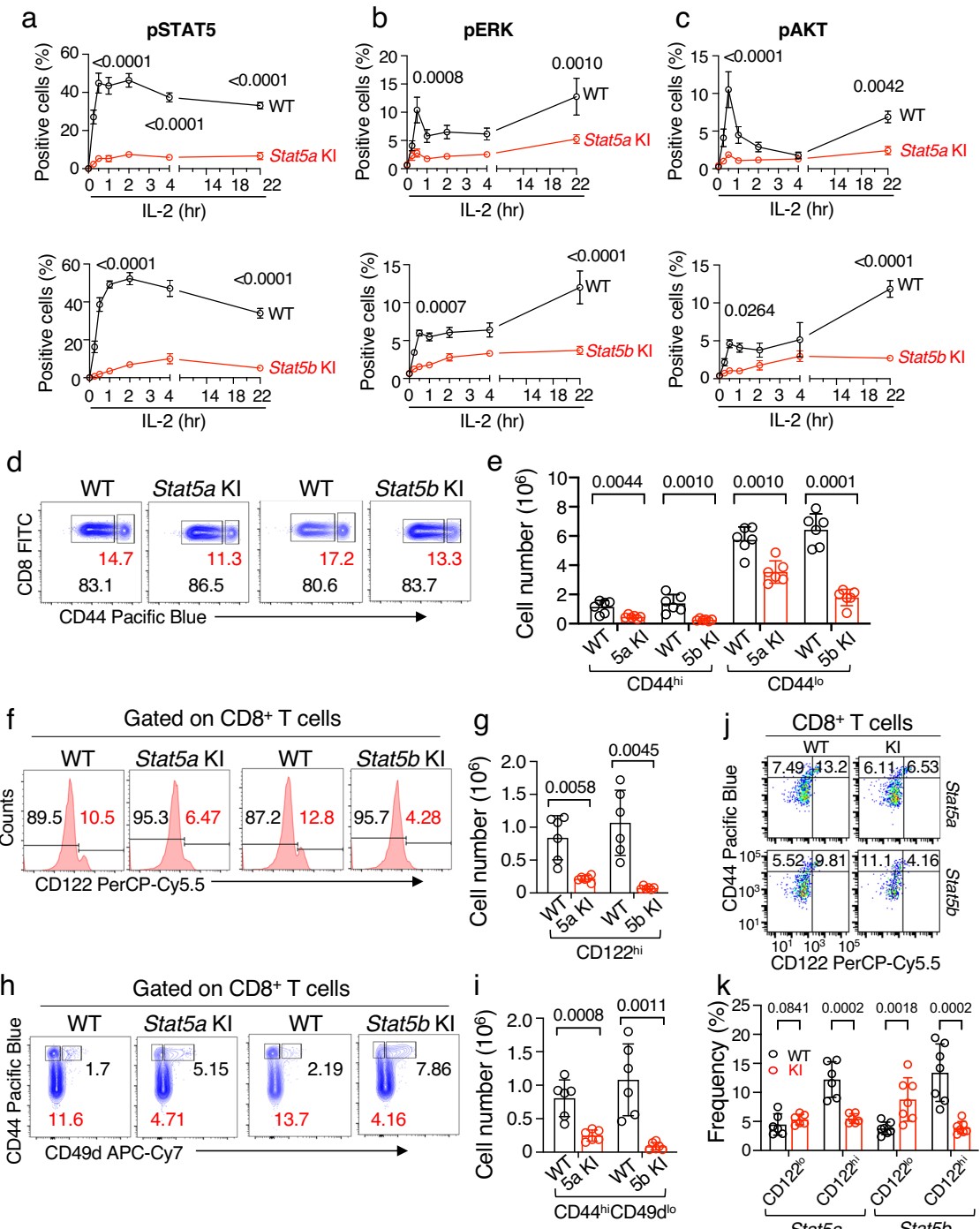

**Fig. 4 | Lower pERK and pAKT levels in STAT5 KI splenic CD8+ T cells stimulated by IL-2, with fewer CD8+CD44hiCD122hiCD49dlo virtual memory cells in these mice.** Time course experiments of IL-2-activated pSTAT5 (**a**, *n* = 4), pERK (**b**, *n* = 4), and pAKT (**c**, *n* = 4) in WT (black lines), *Stat5a* KI (top graphs, red lines), and *Stat5b* KI (bottom graphs, red lines) CD8+ T cells. Each data point represents 4 biological replicates combined from two independent experiments, and each biological replicate was derived from the combined cells from two *Stat5a* KI mice or three *Stat5b* KI mice. Error bars (SEM) and *p*-values are shown. Representative flow cytometric profiles of CD8+ T cells that were stained with antibodies to CD44 (**d**, *n* = 6), CD122 (**f**, *n* = 6), and CD49d (**h**, *n* = 6). Cell numbers of CD8+ T cells that express CD44 (**e**), CD122 (**g**), and CD44 and CD49d (**i**) in WT (black bars/circles), *Stat5a* KI and *Stat5b* KI (red bars/circles) cells. Also shown are error bars (SEM) and

*p*-values determined by multiple unpaired *t* test using the Holm-Šídák method. **j** Representative flow cytometric profiles of CD8+CD44hiCD122lo and CD8+CD44hiCD122hi memory phenotype T cells in WT, *Stat5a* KI, and *Stat5b* KI mice. **k** Frequency of CD8+ memory phenotype T cells in WT (*n* = 7 for *Stat5a* WT and *Stat5b* WT, respectively), *Stat5a* KI (*n* = 7), and *Stat5b* KI (*n* = 7) mice. The gating strategies for **d**, **f**, **h**, and **j** are shown in Supplementary Fig. 10. The numbers in **f** are MFI values of gated area, and **d**, **h**, and **j** are percentage of gated cells; in graphs **e**, **g**, **i**, and **k**, black and red open bars represent WT and mutants, respectively, and the error bars (SEM) and *p*-values are shown, which were determined by multiple unpaired *t* test using the Holm-Šídák method. Source data for **a**–**c**, **e**, **g**, **i**, and **k** are provided as a Source Data File.

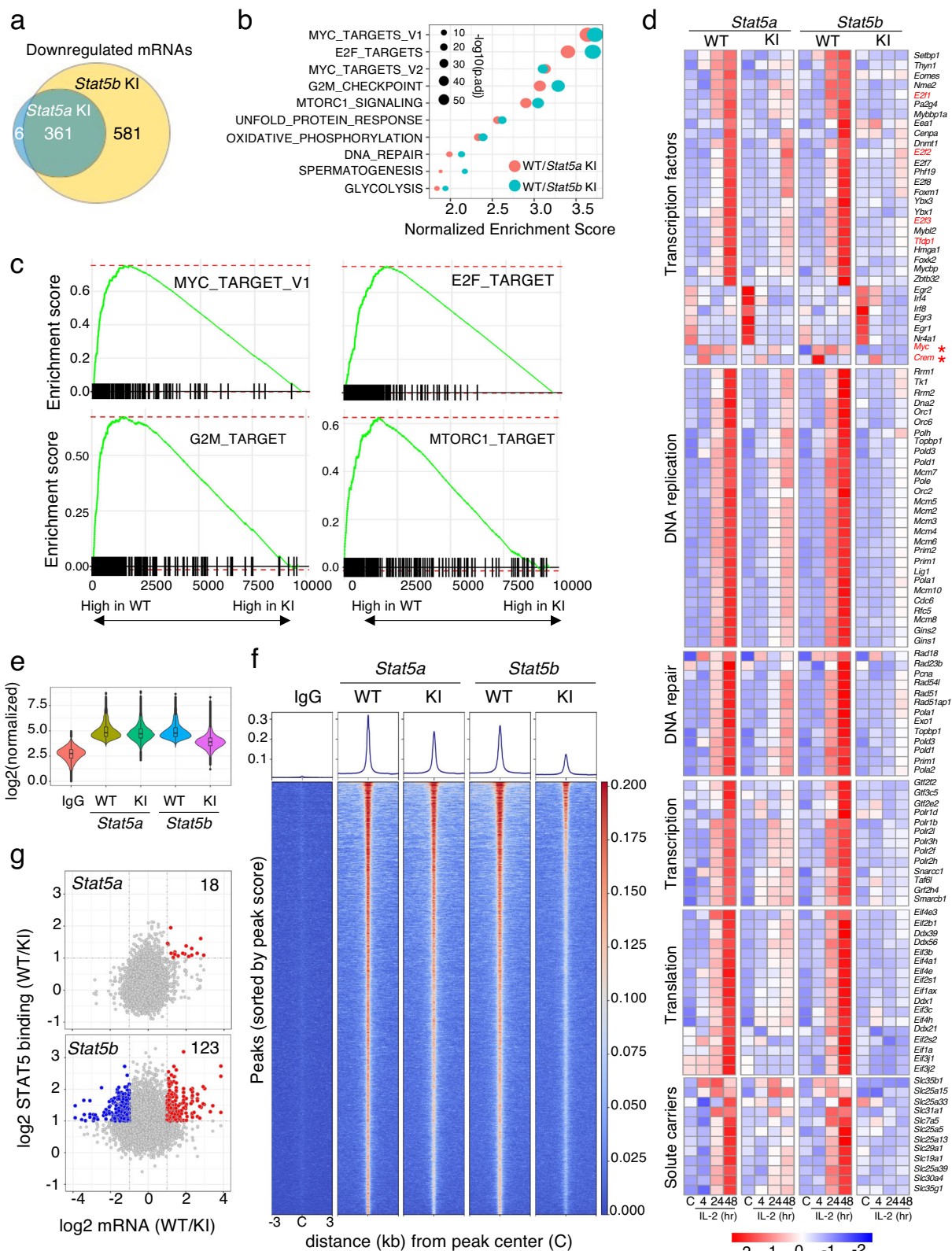

expression of IL-2-target genes at the mRNA and/or protein levels, with their dysregulation collectively contributing to the proliferative defects in *Stat5* KI CD8[+] T cells.

## Discussion

STAT5 plays indispensable roles in mediating the actions of IL-2, IL-7, and IL-15 in the development, expansion, survival, and function of T cells. STAT5 activation by cytokines involves the phosphorylation of a conserved C-terminal tyrosine residue by receptor-associated JAK kinases, which then leads to the formation of STAT dimers, their nuclear translocation, and binding to GAS motifs on target genes, and modulation of target gene expression[4,15]. However, the importance of tyrosine phosphorylation of STAT5A and/or STAT5B in mediating cytokine signaling in vivo has previously not been investigated. Here,

**Fig. 5 | Dysregulation of a broad range of IL-2-induced genes in *Stat5* KI CD8⁺ T cells. a** Venn diagrams of the number of shared and uniquely expressed mRNAs whose expression was lower in *Stat5a* KI and/or *Stat5b* KI CD8⁺ T cells as compared to WT cells. **b** Bubble plot showing top 10 dysregulated genesets shared in *Stat5a* KI (red bubbles) and *Stat5b* KI (bluish green bubbles) CD8⁺ T cells in response to IL-2 stimulation. The -Log10 adjusted *p*-value is indicated by the size of the dots. **c** Shown are MYC_TARGET_V1, E2F_TARGET, G2M_TARGET, and MTORC1_TARGET genesets identified by GSEA. Data from cells stimulated by IL-2 for 48 hr were used to generate **a**–**c**. **d** Heatmaps showing examples of dysregulated mRNAs in *Stat5a* KI and *Stat5b* KI CD8⁺ T cells in response to IL-2, including those for Transcription

factors, DNA replication, DNA repair, Transcription, Translation, and Solute carriers. *Myc*, *E2f1*, *E2f2*, *E2f3*, and *Tfdp1* are highlighted in red. **e** Violin plot of normalized IL-2-induced STAT5 binding peaks in *Stat5a* KI, *Stat5b* KI, and their corresponding WT CD8⁺ T cells. Source data for **e** are provided as a Source Data File. **f** Histographs (top) and heatmap (bottom) comparing IL-2-induced STAT5 binding peaks in WT vs. *Stat5a* KI and WT vs. *Stat5b* KI CD8⁺ T cells. **g** Scatter plots showing log₂(FC) of mRNA (x axis) induced by IL-2 and log₂(FC) of STAT5 binding (y axis) in WT versus *Stat5a* KI (top panel) and WT versus *Stat5b* KI (bottom panel) CD8⁺ T cells. Blue dots show low mRNA expression in WT cells, and red dots show higher mRNA expression in mutant cells. The number of genes is indicated.

we mutated Y694 in STAT5A and Y699 in STAT5B in mice and found that the *Stat5* KI mice unexpectedly had significantly fewer CD8⁺ T cells as compared to those reported in *Stat5a* KO or *Stat5b* KO mice[24–26,30]. A potential explanation for the more severe defects in *Stat5a* KI and *Stat5b* KI mice than in the corresponding KO mice comes from our finding that WT STAT5A and STAT5B, but not mutant STAT5 proteins, form heterodimers in response to IL-2 stimulation, implicating pYSTAT5A:pYSTAT5B heterodimers in regulating STAT5 target gene expression. This suggests that tyrosine-mutated STAT5A and/or STAT5B may act as dominant-negative mutants: the SH2 domain of a tyrosine-mutated STAT5 protein should be able to serve as a docking site for the phosphotyrosine of a WT STAT5 protein, potentially interfering with normal dimer formation between WT STAT5 proteins and function. The fact that similar numbers of STAT5A and STAT5B peptides were identified by mass spectrometry (Supplementary Data 4) indicates that similar levels of each STAT5 protein are expressed in CD8⁺ T cells, suggesting that the dominant role of STAT5B in transducing the IL-2 signal in these cells is intrinsic to the protein rather than due to differences in STAT5A and STAT5B expression.

IL-2 and IL-15 play vital roles in the homeostatic proliferation of CD8⁺ T cells in vivo[41], and IL-2 can robustly expand T cells in vivo and in vitro, including CAR-T cells in cancer immunotherapy[79]. Induction of Myc expression by mitogenic stimuli, including IL-2[18,69,70,80], is important[62], with Myc playing roles in early fate decision of CD8⁺ T cells[81]. We found that both pYSTAT5A and pYSTAT5B are required for optimal IL-2-induced Myc expression, and the lack of pYSTAT5 (particularly pYSTAT5B) prevented normal induction of *Myc* and MYC-target genes, including *E2f*-family genes, which are required for cell cycle entry and progression and cell proliferation. *Crem*, which was the other transcription factor gene rapidly induced by IL-2, bound STAT5 and thus may be a direct STAT5 target gene, whereas no apparent STAT5 binding was detected by ChIP-seq at the *E2f1*, *E2f2*, *E2f3*, and *Tfdp1* loci, indicating that they are not directly regulated by STAT5.

IL-2Rβ and IL-2Rγ are the key receptor chains required for transducing IL-2 signals. Both of these proteins are rapidly internalized and degraded following IL-2 binding[82,83]; thus, de novo synthesis of IL-2Rβ and IL-2Rγ[70] is essential to ensure and maintain a sustained IL-2 response. IL-2-induced expression of IL-2Rβ and IL-2Rγ in freshly isolated CD8⁺ T cells correlated with a rapid and sustained pSTAT5 signal for at least 22 hr. The lower IL-2-induced expression of both IL-2Rβ and IL-2Rγ in mutant cells presumably contributes to their defective IL-2 response, at least in part explaining the diminished activation of pERK and pAKT. These findings underscore the important role for a positive feedback loop of IL-2→IL-2R→pYSTAT5→IL-2R expression in ensuring a sustained IL-2 response. The importance of pYSTAT5 is further underscored by the significantly lower expression of a number of effector molecules in *Stat5* KI CD8⁺ T cells, including IFNγ, TNF, GZMA, GZMB, PRF1, CASP3, CASP7, as well as transcription factors TBX21 and EOMES, which are important for the expression of IL-2Rβ and the development of memory CD8⁺ T cells[44,84].

Phosphorylation of RB by CCND/CDK4/6 at the early G1 phase and CCNE-CDK2 at the late G1 phase are key steps for the release of pRB from the E2F-DP1 complex and the activation of transcription of E2F-target genes. In addition, CDK7, a key component of CDK activating

kinase (CAK, CDK7-CCNH-MAT1), not only plays a role in the activation of cell cycle progression-related CCNs/CDKs, including CCND/CDK4/6 and CCNE/CDK2, but it is also a component of TFIIH and thereby plays a key role in the initiation of transcription by RNA Pol II[73,74]. Consistent with their important roles in IL-2-induced proliferation, the levels of pRB, Myc, E2F, CCND1, CCNA2, CCNE, CDK2, CDK4, CDK6, and CDK7 were all significantly lower in *Stat5* KI CD8⁺ T cells, providing molecular insights into the basis of the defective IL-2-induced cell cycle progression of *Stat5* KI cells. Indeed, a gene set enrichment analysis of our transcriptomics data revealed that the top 10 genesets that were significantly dysregulated in *Stat5* KI mutant cells all are related to cell cycle entry and cell cycle progression; these include Myc- and E2F-target genes and genes involved in G2M checkpoint, MTORC1 signaling, oxidative phosphorylation, spermatogenesis, DNA repair, glycolysis, and the mitotic spindle. Moreover, our mass spectrometric analysis revealed a number of differentially expressed proteins in *Stat5a* KI CD8⁺ T cells, whereas their mRNA levels were not significantly altered, indicating defective post-transcriptional regulation in the mutant cells as well.

In summary, by investigating the molecular basis of the defective proliferation in *Stat5* KI CD8⁺ T cells using cell cycle analysis as well as transcriptomic and proteomic analyses, we have uncovered a broad range of genes that are directly and/or indirectly regulated by pYSTAT5 at the mRNA and/or protein levels and are vitally important for a wide range of biological processes. These include the IL-2 receptor components, IL-2Rβ and γ_c, and many proteins that are critically important for DNA replication, transcription, translation, DNA repair, metabolism, nuclear import and export, and protein degradation, as well as genes encoding for solute carriers and ABC transporters (Fig. 8). Our findings provide a valuable resource of in-depth information of IL-2-regulated gene expression at both the mRNA and protein levels, with new insights into IL-2 signaling and IL-2 target genes, potentially revealing additional means to modulate and fine-tune the IL-2 response.

## Methods

### Generation of STAT5A^Y694F knockin (KI) and STAT5B^Y699F KI homozygous mice and STAT5^Y694F/Y699F double KI (DKI) heterozygous mice

The STAT5A^Y694F and STAT5B^Y699F mouse lines were generated by CRISPR/Cas9 methodology[85]. Briefly, single guide RNAs (sgRNAs) for *Stat5a* and *Stat5b* (Supplementary Table 1) were designed to cut near the mutation site of each gene. The sgRNA sequences were cloned into a sgRNA vector using OriGene's gRNA Cloning Services (Rockville, Maryland) and used as templates to synthesize sgRNAs using the MEGAshortscript T7 Kit (Life Technologies, AM1354). Cas9 mRNA was in vitro transcribed from plasmid MLM3613 (Addgene #42251) using the mMESSAGE mMACHINE T7 Ultra Kit (Life Technologies, AM1345). Single strand donor oligonucleotides for *Sta5a* and *Stat5b* (Supplementary Table 1), respectively, were used for introducing the point mutations. Besides the desired nucleotide changes (Supplementary Fig. 1a, highlighted red letters) for changing the Tyr to Phe, three silent nucleotides (lower case letters in red) that do not result in amino acid changes but decrease the likelihood of Cas9 continuing to cut the DNA after the donor DNA is knocked in, were also included in the donor oligonucleotides. For making each

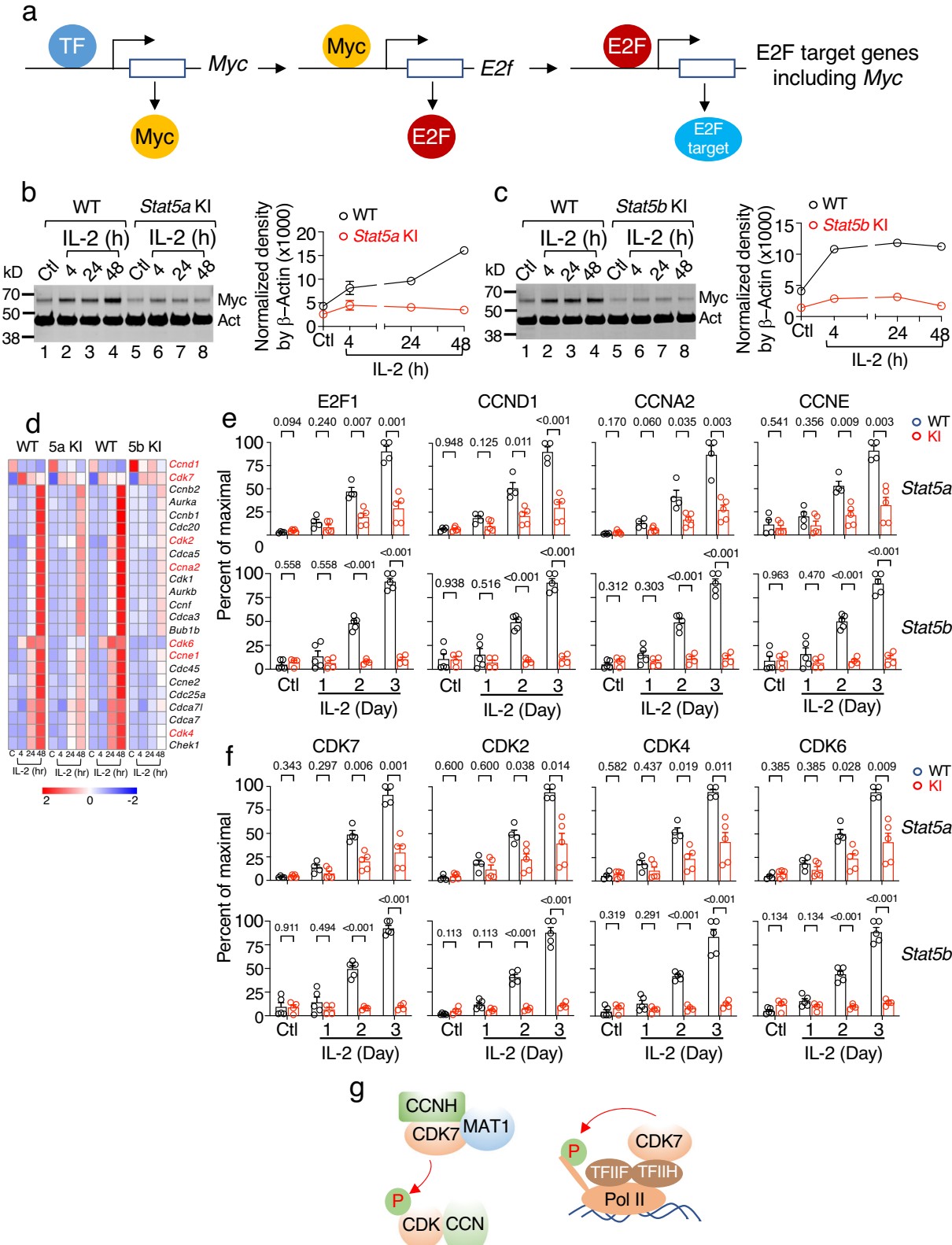

mouse line, the sgRNA (50 ng/μl) and its corresponding donor oligonucleotides (100 ng/μl) were co-microinjected with Cas9 mRNA (100 ng/μl) into the cytoplasm of zygotes collected from B6D2F1 mice (JAX Stock No: 100006). Injected embryos were cultured in M16 medium in a 37 °C incubator with 6% $CO_2$. Those embryos reached 2-cell stage of development were implanted into the oviducts of pseudopregnant surrogate mothers. The *Stat5a/Stat5b* DKI mouse

line (STAT5A[Y694F]/STAT5B[Y699F]) was generated by sequentially targeting the two genes, which are located on mouse chromosome 11 in close proximity. Zygotes were collected from mice harboring the STAT5B[Y699F] mutation and were then microinjected with *Stat5a* sgRNA and *Stat5a* donor oligos following the procedures described above. Founder mice for each KI mouse line were identified by PCR followed by Sanger sequencing and backcrossed to C57BL/6 J

**Fig. 6 | Decreased expression of Myc, E2F1, CCND1, CCNA2, CCNE, CDK2, CDK4, CDK6, and CDK7 in IL-2-stimulated *Stat5* KI CD8⁺ T cells. a** Schematic showing that expression of Myc leads to upregulation of E2F and E2F target genes, including Myc itself.) **b** and **c** Showing representative western blots of Myc expression from two experiments in freshly isolated WT and *Stat5a* KI (**b**) and WT and *Stat5b* KI (**c**) CD8⁺ T cells stimulated by IL-2 for indicated time points. Line graph on the right of each panel shows Myc levels normalized by blotting with anti-β-actin (**b** for *Stat5a* KI and **c** for *Stat5b* KI samples). Source data for **b** and **c** are provided as a Source Data File. **d** Heatmap showing selected cell-cycle related mRNAs whose expression was attenuated in *Stat5a* KI and *Stat5b* KI CD8⁺ T cells in response to IL-2 stimulation. **e** Percentage of cells expressing E2F1, CCND1, CCNA2, and CCNE, as indicated, in WT (black open bars), *Stat5a* KI (top panels, red open bars) and *Stat5b* KI (bottom panels, red open bars) CD8⁺ T cells simulated by IL-2; n = 4 for *Stat5a* WT, n = 5 for

*Stat5a* KI, n = 4 for both *Stat5b* WT and *Stat5b* KI. **f** Percentage of cells expressing CDK7, CDK2, CDK4, and CDK6 in WT (black open bars), *Stat5a* KI (top panels, red open bars) and *Stat5b* KI (bottom panels, red open bars) CD8⁺ T cells simulated by IL-2. **e, f** Shown are combined data from two independent experiments using *Stat5a* CD8⁺ T cells (top panels) and *Stat5b* CD8⁺ T cells (bottom panels). Each data point represents 4 or 5 biological replicates (n = 4 for *Stat5a* WT, n = 5 for *Stat5a* KI, n = 5 for *Stat5b* WT and n = 4 for *Stat5b* KI) and each biological replicate was derived from the combined cells from three *Stat5b* KI mice. Also shown are error bars (SEM) and *p*-values determined by multiple unpaired *t* test using the Holm-Šídák method. Source data for **b**, **c**, **e**, and **f** are provided as a Source Data File. **g** Schematic showing CDK7 as a key component of CDK activating kinase (CAK) that phosphorylates DNA polymerase II to regulate gene transcription.

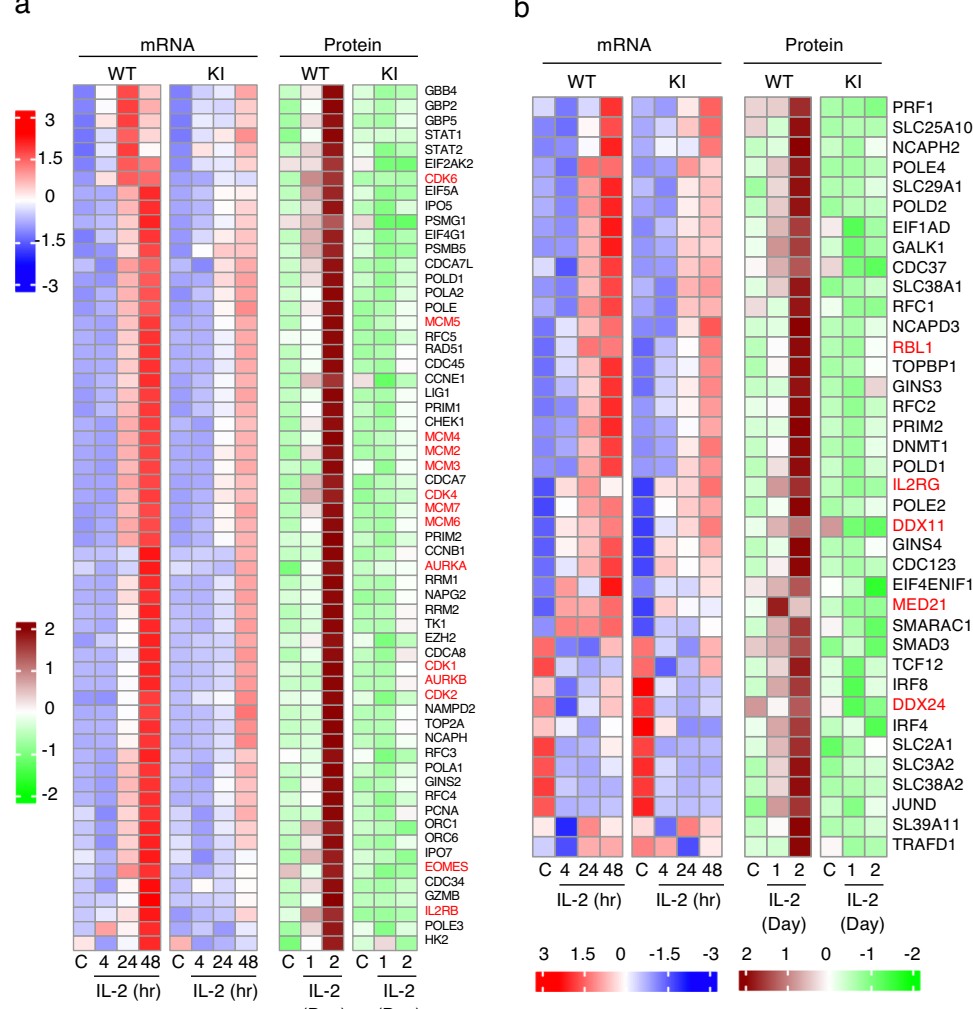

**Fig. 7 | Dysregulated protein expression in *Stat5a* KI CD8⁺ T cells identified by global proteomics analysis. a** Heatmap showing mRNAs and proteins similarly dysregulated in *Stat5a* KI cells identified by RNA-seq and MS. CDK1, CDK2, CDK4, CDK6, MCM2, MCM3, MCM4, MCM5, MCM7, AURKA, AURKB, IL2RB, and EOMES are highlighted in red. **b** Heatmap of selected mRNAs that were not

dysregulated or only modestly dysregulated (FC < 2) in *Stat5a* KI CD8⁺ T cells, but whose corresponding protein levels were significantly lower in the KI cells as compared to WT cells in response to IL-2 stimulation. RBL1, IL2RG, DDX11, MED21, and DDX24 are highlighted in red.

(JAX 000664) background for 6 generations before using in experiments, and we used the littermate WT mice for each line as the appropriate control.

## Mice
Both female and male mice from 8 to 24 weeks old were used for the experiments. The mice were housed in specific pathogen-free mouse

facilities at the National Institutes of Health. The bone marrow, thymi, and spleens of mice were collected after the mice were euthanized by carbon dioxide inhalation. The WT and mutant mice were co-housed and were generated by mating heterozygous x heterozygous mice. All mouse protocols were approved by the National Heart, Lung and Blood Institute Animal Care and Use Committee, and experiments followed NIH guidelines for using animals in intramural research.

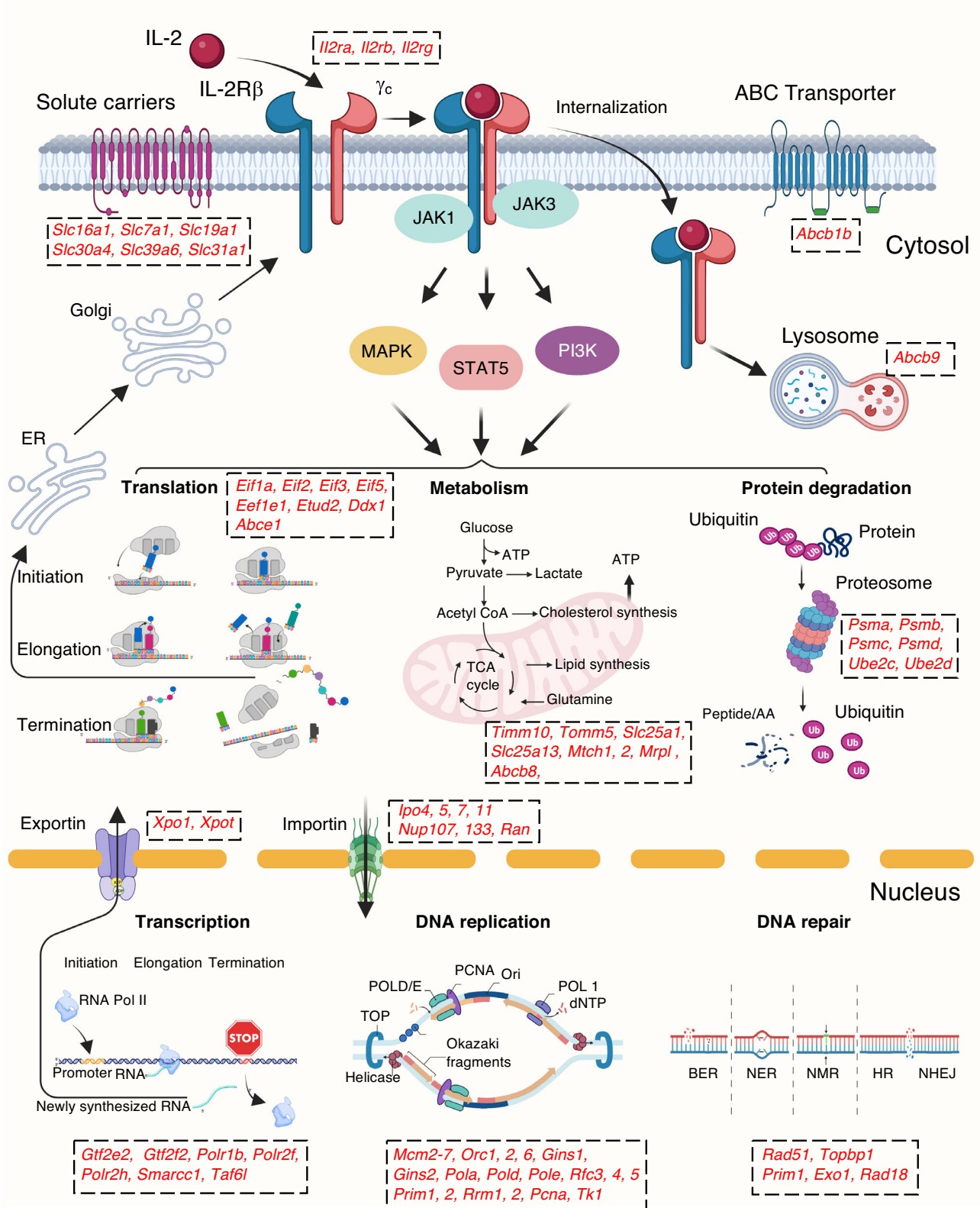

**Fig. 8 | Molecular basis of defective IL-2-induced proliferation in *Stat5* KI CD8⁺ T cells.** Schematic showing that activation of MAPK, STAT5, and PI3K pathways by IL-2 in freshly isolated WT CD8⁺ T cells induces increased expression of genes at the mRNA and/or protein levels that are important for cell-cycle progression and proliferation, including IL-2Rβ, IL-2Rγ, solute carriers, transporters, nuclear pore components, as well as molecules involved in mitochondrial metabolism, transcription, DNA replication, DNA repair, translation, and protein degradation. The protein expression of IL-2-induced genes in these processes was defective in *Stat5* KI cells. The genes shown in red were significantly induced by IL-2 (FC ≥ 2) in WT cells but had markedly diminished induction in *Stat5* KI cells. Abbreviations: BER, base excision repair; NER, nucleotide excision repair; MMR, mismatch mediated repair; HR, homologous recombination, and NHEJ for non-homologous end-joining. This figure was generated using BioRender (BioRender agreement number AP26LFT6A4).

## Cell purification and culture

Splenic total T cells were purified using EasySep Mouse T cell Isolation Kit (STEMCell Technologies, 19851) from WT and mutant mice, cultured in complete RPMI 1640 medium (supplemented with 100 U/ml penicillin and streptomycin, 50 μM β-mercaptoethanol, 2 mM glutamine, and 10% FBS) with plate-bound anti-CD3 (2 μg/ml, 145-2C11, BioXCell) and soluble anti-CD28 (1 μg/ml, 37.51, BioXCell) for 3 days, the cells were then washed 3 times with complete RPMI 1640 medium, and cultured in complete RPMI 1640 medium without cytokines overnight. Splenic CD8[+] T cells were purified using the EasySep Mouse CD8[+] T cell Isolation Kit (STEMCELL Technologies, 19853) and cultured in complete RPMI 1640 medium either without or with 500 IU/ml recombinant human IL-2 (Biological Resources Branch Preclinical Biologics Repository, NCI).

## Flow cytometric analyses

For cell surface marker staining, 10[6] cells in 100 μl PBS containing 0.5% BSA were stained with fluorescent-labelled or unlabeled monoclonal antibodies (BD Biosciences, San Jose, CA, BioLegend Inc., San Diego, CA, Cell Signaling Technology, Danvers, MA, or ThermoFisher Scientific, Grand Island, NY) for B220 (RA3-6B2), CD3ε (145-2C11), CD4 (GK.5), CD8a (53-6.7), CD44 (IM7), CD49d (R1-2), RB1 (1F8), pRB (S807/811, D20B12), CD25 (7D4), CD25 (3C7), CD122 (TM-b1), DX5 (DX5), NK1.1 (PK1.36), NKp46 (CD335), CD49b (DX5), B220 (RA3-6B2), E2F1 (JJ092-02, MA5-32476) CCND1 (SP4, MA5-14512), CCNA2 (SD2052, MA5-32353), CCNE (HE12, MA5-14336), CDK2 (1A6, MA5-17052), CDK4 (DCS-31, AHZ0202), CDK6 (K6.83 (DCS-83), AHZ0232), CDK7 (JJ203-01, MA5-32434), Alexa Fluor 405 Goat anti-Rabbit IgG (H + L, A31556), Alexa Fluor 488 Goat anti-Rabbit IgG (Heavy chain, A27034), Alexa Fluor 647 Goat anti-Rabbit IgG (Heavy chain, A27040), Alexa Fluor 750 Goat anti-Rabbit IgG (H + L, A21039), Alexa Fluor 488 Goat anti-Mouse IgG (H + L, A28175), PE F(ab')2-Goat anti-Mouse IgG (H + L, 12-4010-82), Alexa Fluor 647 Goat anti-Mouse IgG (H + L, A21236); for Foxp3 staining, the cells were fixed and permeabilized before staining with PE-labelled FOXP3 monoclonal antibody (FJK-16s, ThermoFisher, 12-5773-82) or Alexa647-FOXP3 monoclonal antibody (BioLegend, 320014) (Supplementary Table 2). To fix and permeabilize cells for staining with pY694STAT5 (47/STAT5 pY694, BD, 612599), pERK (197G2, CST, 13214 S), and pAKT (D9E, CST, 5315 S), the cells were fixed using 1.5% paraformaldehyde or 4% formaldehyde at room temperature for 15 min, then permeabilized with cold 100% methanol on ice for 20 min. Data were acquired using a FACSCanto II flow cytometer (BD Immunocytometry Systems) and analyzed using FlowJo (v10.9.0, BD).

Ex vivo experiment. For ex vivo experiments, 30 μg of MSA-conjugated IL-2 was injected i.p. into 6–8 month-old mice on days 0, 3, and 6. Total spleen cells were isolated on day 7, red blood cells were lysed, and total splenocytes were counted and stained with anti-CD3, anti-CD122, anti-NK1.1 for NK cells; anti-CD3, anti-CD8, anti-CD44, anti-CD122 for CD8[+] T cells; and anti-CD3, anti-CD4, anti-CD25, and anti-Foxp3 for CD4[+] T cells. The data were aquired on a FACSCanto II flow cytometer and analyzed using FlowJo.

## Cell proliferation assay

To monitor cell division, freshly isolated CD8[+] or total splenic T cells from WT or tyrosine mutant mice were labeled with CellTracer CFSE Cell Proliferation Kit in PBS (ThermoFisher, C34554) as previously described[18]. 1 × 10[6]/ml CFSE-labeled cells were then cultured in the presence of 500 IU/ml recombinant human IL-2, or in the presence of 2 μg/ml plate-bound anti-CD3 and 1 μg/ml soluble anti-CD28. CD8[+] T cell division was monitored by CFSE dilution using flow cytometry on days 3, 4, and 5 for IL-2 stimulation and on days 2 and 3 for TCR stimulation.

## Cell cycle analysis

Cell cycle analysis was performed using Click-iT Plus EdU Alexa Flour 488 Kit (ThermoFisher, C10633) and FxCycle Violet Ready Flow Reagent (ThermoFisher, R37166) per the manufacturer's recommendation. Briefly, purified CD8[+] T cells were cultured in complete RPMI 1640 medium with 500 IU/ml IL-2, and at indicated time point, the cells were labeled with EdU at 10 μM final concentration at 37 °C for 1 hr. After washing and performing the Click reaction, the cells were washed again, permeabilized, and stained with FxCycle Violet. Data were acquired on a FACSCanto II flow cytometer and analyzed by Flowjo software.

## Immunoprecipitation and western blotting

For co-IP and IP experiments, preactivated total T cells were washed with complete RPMI 1640 medium and cultured in complete RPMI 1640 medium without cytokines overnight. Equal numbers of cells were cultured without or with 500 IU/ml of IL-2 for 20 min. Total cell lysates were prepared using high salt NP40 lysis buffer (10 mM Tris, pH 7.5, 0.3 M NaCl, 0.5% NP40, cOmplete Protease Inhibitor Cocktail tablet, and PhosSTOP tablet) on ice for 20 min and then cleared by centrifugation at 14,000 rpm for 15 min. The cleared lysates were then subjected to immunoprecipitation using PureProteome Protein A/G Mix Magentic Beads (Millipore Sigma LSKMAGAG) and rabbit anti-STAT5A (R1216) or anti-STAT5B (R1219)[2] antisera at 4 °C overnight. After washing with NP40 lysis buffer, immunoprecipitation products were denatured in LDS sample buffer with 2% β-mercaptoethanol and separated on a 4–12% NuPage Bis-Tris gel (ThermoFischer, NP0322), transferred on an Immobilon®-LF PVDF membrane (Millipore Sigma, IPFL10100), blotted using mouse monoclonal anti-STAT5A (R&D, MAB2174), anti-STAT5B (R&D, MAB1584), and anti-pYSTAT5 antibodies (CST, 9356), respectively (Supplementary Table 3). After washing, the membrane was visualized by using IRDye 680RD goat anti-mouse IgG secondary antibody (LI-COR Biosciences, 926-68070) on an Odyssey CLx Imaging system (LI-COR Biosciences) (Supplementary Table 3).

For western blot analysis of Myc, purified splenic CD8[+] T cells were cultured in complete RPMI 1640 medium either with 500 IU/ml recombinant human IL-2 for 4 hr, 1 day, or 2 days. Total cell lysates prepared using high salt NP40 lysis buffer with cOmplete ULTRA Tablets (Millipore Sigma, 05892970001) and PhosSTOP (Millipore Sigma, 04906837001) from cells without IL-2 or the cells were stimulated with IL-2 at predetermined time point. After denaturing in LDS sample buffer with 2% β-mercaptoethanol, the lysates were separated on a 4–12% NuPage gel, transferred on Immobilon®-LF PVDF membrane (Millipore Sigma IPFL10100), blotted using rabbit monoclonal anti-Myc (CST, 18583) and mouse monoclonal anti-β-actin (CST, 3700) antibodies (Supplementary Table 3), and visualized using IRDye 680RD donkey anti-mouse IgG secondary antibody (LI-COR Biosciences, 926-68072) or IRDye 800CW donkey anti-rabbit IgG secondary antibody (LI-COR Biosciences, 926-32213) (Supplementary Table 3) on an Odyssey CLx Imager.

## Total RNA isolation

The purified CD8[+] T cells were cultured in the presence of 500 U/ml of recombinant human IL-2, and then collected at 0, 4, 24, and 48 hr. The cells were washed with PBS once, lysed in TriReagent (Zymo Research, Irvine, CA), and total RNA was isolated using Direct-zol RNA MicroPrep Kit (Zymo Research, IR2060).

## RNA-seq library preparation and sequencing

RNA-seq libraries were prepared as previously described[19] using 100 ng of total RNA from splenic CD8[+] T cells and the KAPA mRNA HyperPrep Kit (KR1352, KAPA Biosystems, Wilmington, MA), and each library was indexed using NEXflex DNA Barcodes-24 (BIOO Scientific, Austin, TX). Barcoded libraries were separated on 2% E-Gel, 250–400 bp fragments were purified, quantified on Qbit (Invitrogen), and equal amounts of each library were pooled and sequenced on an Illumina NovaSeq platform (Illumina, San Diego, CA).

## ChIP-Seq library preparation and sequencing

ChIP-seq libraries were prepared as previously described[19] with some modification. Briefly, freshly isolated splenic CD8+ T cells from *Stat5a* KI, *Stat5b* KI, and WT littermate mice were either not treated or treated with 500 IU/ml of IL-2 for 4 h, and cross-linked with 1% formaldehyde (methanol-free, Pierce, Rockford, IL) at room temperature for 10 min. After sonication at 30" ON and 30" OFF for 5 times using Bioruptor Pico (Diagenode), fragmented chromatin equivalent to 7 million cells were immunoprecipitated with control rabbit IgG (3900 S, Cell Signaling Technology) or anti-STAT5A/B antibody (Ab194898, Abcam) (Supplementary Table 4), and 20 µl of Magna ChIP Protein A + G Magnetic Beads (16–663, Millipore, Billerica MA). ChIP-Seq DNA libraries were prepared using KAPA Hyper Prep Kit (KR0961, KAPA Biosystems, Wilmington, MA) and NEXflex DNA Barcodes-24 (NOVA-514103, BIOO Scientific, Austin, TX), and the indexed libraries were pooled and sequenced on an Illumina NovaSeq platform.

## RNA-seq analysis

Sequenced reads (50 bp, single end) were obtained with the Illumina CASAVA pipeline and mapped to the mouse genome (mm10/GRCm38) using TopHat 2.0.11. Only uniquely mapped reads were retained. RefSeq gene database was downloaded from the UCSC genome browser for RNA-seq analysis. Raw counts that fell on exons of each gene were calculated and normalized by using RPKM (Reads Per Kilobase per Million mapped reads). Differentially expressed genes were identified with the R Bioconductor package "edgeR", and expression heat maps were generated with the R package "pheatmap". For GSEA analysis, RNA-seq based gene expression data were compared with molecular signature gene sets using R package "fgsea".

## ChIP-Seq analysis

Sequenced reads mapped to the mouse genome (mm10/GRCm38) using Bowtie 2.2.6. Only uniquely mapped reads were retained. The mapped outputs were converted to browser-extensible data files, which were then converted to binary tiled data files (TDFs) using IGVTools 2.4.13 for viewing on the IGV browser (http://www.broadinstitute.org/igv/home). TDFs represent the average alignment or feature density for a specified window size (i.e., 20 bp sliding window) across the genome. The reads were shifted 100 bp from their 5' starts to represent the center of the DNA fragment associated with the reads. Peaks were identified by using Model-based Analysis of ChIP-Seq (MACS 1.4.2) compared to IgG control library. Normalized peak intensities were calculated by HOMER (Hypergeometric Optimization of Motif EnRichment, v4.11). Motif analysis was done using Motif-based sequence analysis tools MEME 5.4.1.

## LC/MS analysis

For trypsin/LysC digestion and TMTpro labeling, cells were lysed from the frozen cell pellets in 100 µl EasyPep Lysis buffer (Thermo Fisher PN A45735) and total cell lysates were treated with 1 µl universal nuclease (Thermo PN 88700). Protein concentration was determined by the BCA method, and 10 µg of protein from each sample were subjected to reduction and alkylation at 25 °C in the dark for 1 hr and then digested at 37 °C overnight for a total of 19.5 hrs. The peptides from each sample were then labeled using TMTpro 18-plex label reagents (Thermo PN A52045) at 25 °C for 1 hr. The excess TMTpro was quenched with 50 µl of 5% hydroxylamine, 20% Formic acid for 10 min, the samples were then combined, cleaned using EasyPep mini columns, and eluted peptides were dried in speed-vac.

The TMTpro labeled peptides were fractionated by High-pH reverse phase liquid chromatography using a 150 mm × 3.0 mm Xbridge Peptide BEMTM 2.5 µm C18 column (Waters, MA) and a Waters Acquity UPLC system with a fluorescence detector (Waters,

Milford, MA) at 0.35 ml/min. After drying, the peptides were reconstituted in 50 µl of mobile phase A (10 mM ammonium formate, pH 9.3), applied on a 150 mm × 3.0 mm Xbridge Peptide BEMTM 2.5 µm C18 column (Waters, MA), and eluted in mobile using gradient elution of 10–50% phase B (10 mM ammonium formate, 90% ACN) from 1.5–60 min followed by 50–90% phase B for 60–65 min. Sixty-five fractions were collected, and these were consolidated into 12 pools based on the chromatogram intensity and vacuum centrifuged to dryness. Each fraction was resuspended in 60 µl of 0.1% FA and 15 µl was analyzed using a Dionex U3000 RSLC in front of a Orbitrap Lumos (Thermo) equipped with an EasySpray ion source. Solvent A was 0.1% FA in water and Solvent B was 0.1% FA in 80% ACN. Loading pump consisted of Solvent A and was operated at 7 µl/min for the first 6 min of the run then dropped to 2 µl/min when the valve was switched to bring the trap column (Acclaim PepMap 100 C18 HPLC Column, 3 µm, 75 µm I.D., 2 cm, PN 164535) in-line with the analytical column EasySpray C18 HPLC Column, 2 µm, 75 µm I.D., 25 cm, PN ES902). The gradient pump was run at a flow rate of 300 nl/min, and each run used a linear LC gradient of 5–7% B for 1 min, 7–30% B for 134 min, 30–50% B for 35 min, 50–95% B for 4 min, holding at 95% B for 7 min, then re-equilibration of analytical column at 5% B for 17 min. MS acquisition employed the TopSpeed method with a 3 s cycle time at spray voltage of 1800V and ion transfer temperature of 275 °C. MS1 scans were acquired in the Orbitrap with resolution of 120,000, AGC of 4e5 ions, and max injection time of 50 ms, mass range of 400–2000 m/z, and MS2 scans were acquired in the Orbitrap with a resolution of 50,000, AGC of 1.25e5, max injection time of 86 ms, HCD energy of 38%, isolation width of 0.7 Da, intensity threshold of 2.5e4 and charges 2-5 for MS2 selection. Advanced Peak Determination, Monoisotopic Precursor selection (MIPS), and EASY-IC for internal calibration were enabled and dynamic exclusion was set to a count of 1 for 15 sec.

## Database search and post-processing analysis of LC/MS data

MS files were searched with Proteome Discoverer 2.4 using the Sequest node. Data was searched against the Uniprot Mouse database from Feb 2020 using a full tryptic digest, 2 max missed cleavages, minimum peptide length of 6 amino acids and maximum peptide length of 40 amino acids, MS1 mass tolerance of 10 ppm, MS2 mass tolerance of 0.02 Da, fixed modifications for TMTpro (+ 304.207) on lysine and peptide N-terminus and carbamidomethyl (+ 57.021) on cysteine, and variable oxidation on methionine (+ 15.995 Da). Percolator was used for FDR analysis and TMTpro reporter ions were quantified using the Reporter Ion Quantifier node and normalized using the total peptide intensities of each channel. The Log2FC (Median of groups) and *p*-values (ANOVA) were calculated with PD2.4 software and only proteins with quantitative values in 8 or more samples were included for the analysis. FDR was set to ≤ 1%. Proteins with *p*-value ≤ 0.05 and Log$_2$(FC) cutoffs of ≥ 0.4 were considered differentially expressed.

## Statistical analysis

Numerical data are presented as mean ± SD or mean ± SEM. GraphPad Prism software (v10.0.3) was used to analyze the data. Multiple unpaired *t* test using two-stage step-up method of Benjamini, Krieger and Yekutieli, or Two-way ANOVA multiple comparisons using the Šídák hypothesis test, or multiple unpaired *t* test using the Holm-Šídák method were performed as indicated in the figure legends. The level of statistical significance was set at *p* < 0.05. The number of experiments and mice used are indicated in the figure legends.

## Reporting summary

Further information on research design is available in the Nature Portfolio Reporting Summary linked to this article.

## Data availability

The mass spectrometry data are uploaded in the Mass Spectrometry Interactive Virtual Environment (MassIVE) with accession MSV000093320, https://massive.ucsd.edu/ProteoSAFe/static/massive.jsp?redirect=auth, and the FTP download link is ftp://massive.ucsd.edu/v06/MSV000093320/. The RNA-seq and ChIP-seq data are available at the Gene Expression Omnibus database repository under accession number GSE247343. All data are included in the Supplementary Information or are available from the authors, as are unique reagents used in this Article. The raw numbers for charts, graphs, and blots are available in the Source Data File whenever possible. Source data are provided with this paper.

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

## Acknowledgements

This work was supported by the Division of Intramural Research, NHLBI. Next-generation sequencing was performed in the NHLBI DNA-sequencing core, mutant mice were generated in the NHLBI transgenic core, histology experiment was performed in the NHLBI pathology core,

and mass spectrometry analysis was perfomed in NCI protein characterization laboratory. We thank Dr. David L. Levens (NCI) for critical comments on the manuscript, and Dr. Ning Du for help with immunophenotyping experiments.

## Author contributions

J.-X.L and W.J.L designed and supervised the project, interpreted the data, and worte the manuscript. J.-X.L, M.G., and R.S. performed experiments, C.L and J.-X.L. designed oligos to generate mutant mouse lines, and T.G contributed animal experiments. R.H. performed mass spectrometry experiment and R.H. and T.A. analyzed and interpreted the data. Z.Y. analyzed the histology data. P.L. analyzed, interpreted, and visaulized RNA-seq, ChIP-seq, and mass spectrometry data. J.-X.L. and W.J.L. edited and finalized the manuscript with input from the co-authors.

## Funding

## Competing interests

The authors declare no competing interests.
