## [Peer Review File · Nature Communications]

Tyrosine phosphorylation of both STAT5A and STAT5B is necessary for maximal IL-2 signaling and T cell proliferation.REVIEWER COMMENTS

Reviewer #1 (T cell activation) (Remarks to the Author):

The manuscript by Lin et al. addresses the role of STAT5 phosphorylation in vivo. They generated knock-in mice with mutated phosphorylation sites in STAT5A and STAT5B, respectively. They observed reduced numbers of lymphocytes in the homozygous knock-in mice with the strongest phenotype in CD8+ T cells (numbers and phenotype of these cells in vivo, response to IL-2 ex vivo). The subsequent omics experiments (RNAseq, proteomics, ChIPseq) revealed substantial defects in the IL-2 signaling on the basis of individual genes. This article reveals the importance of STAT5 phosphorylation in primary cells, documents the dependency of antigen-inexperienced memory-like T cells (aka virtual memory) on STAT5 signaling, shows that STAT5 stimulates other IL-2-induced pathways such as ERK and AKT (probably indirectly), suggests that STAT5B might be more active than STAT5A, and last, but not least, provides very useful resources of IL-2 and/or STAT5 target genes, which will be probably heavily used by the community (including my lab).

The experiments are very well planned and executed. The conclusions are supported by the strong experimental evidence. The data are presented in a very clear way. Overall, I have to congratulate the authors on a very nice and appealing work and I recommend it for publication in Nature Communications.

Minor comments:

1. I do not find it necessary to refer to figure details (such as lanes, top/middle panels) in the main text. The figures are self-explanatory and the figure legend might be more appropriate for these detailed explanations, if necessary.

2. Line 118 - there is a comma instead of a point at the end of the sentence.

3. Figure 4a-c - It should be indicated for how long the cells used for these analyses were stimulated with IL-2 (or is it all time points combined?).

4. The authors claim that "(...) antigen naïve virtual memory cells, which require (...) IL-7 for their maintenance" and cite two papers by Akue et al. and Quinn et al. However, these two papers do not address the importance of IL-7 for virtual memory T cells.

Reviewer #2 (IL-2 biology) (Remarks to the Author):

The authors used CRISPR-Cas9 gene editing to generate Stat5a(Y694F) and Stat5b(Y699F) tyrosine mutant knockin (KI) mice, both in homozygous and heterozygous constellation. When analyzing the animals, they observed similar frequencies of CD4⁺ and CD8⁺ thymocytes in homo- and heterozygous KI mice, but lower CD8⁺ number in homozygous KI mice. Total splenic cell counts were lower in all KI mice, resulting in altered counts of T and B cells compared to wildtype (WT) animals. Frequencies and cell counts of CD4⁺ CD25⁺ T cells, CD8⁺ CD122^{hi} T cells and CD8⁺ CD44^{hi} CD122^{hi} CD49d^{lo} T cells were decreased in all KI mouse lines. Compared to Stat5 knockout (KO) mice, more severe defects of T cell counts were observed in KI mice, suggesting a dominant negative effect of the KI constructs. Moreover, all KI mice showed decreased bone marrow and decreased mature NK cell counts compared to WT animals. When comparing IL-2- and TCR-induced proliferation of CD8⁺ T cells, the authors observed all KI T cells proliferated less upon IL-2 stimulation, whereas TCR-induced proliferation was normal. They performed RNA-seq analysis of WT and KI CD8⁺ T cells at different timepoints after IL-2 stimulation and observed several differentially expressed genes (DEGs) in WT and KI cells; the top ten hallmark genesets dysregulated in both KI mutants were related to cell cycle progression (also confirmed by assessment of cell cycle proteins), and expression of known IL-2-inducible genes were reduced in the KI mutants. Finally, using mass spectrometry-based proteomics, the authors examined the abundance of proteins upon IL-2 stimulation of WT and KI mutant cells, identifying 524 differentially expressed proteins, of which 486 were significantly lower in KI mutant cells compared to WT. These proteomics data were consistent with the RNA-seq data of the authors.

Overall, this is a very well written and nicely presented manuscript summarizing a significant body of work, which provides novel insights into the biology of the STAT5 signaling pathway *in vivo* in mice. Below are some points to improve the manuscript.

1) When assessing CD8⁺ T cells of KI mice, the authors used CD8 vs. CD122 staining and gated on CD8⁺ CD122^{hi} T cells, which are known to be stimulated by IL-2 and IL-15. However, gating on CD8⁺ CD122^{hi} T cells is not very easy, as also exemplified by the dot plots provided in this manuscript in Fig. 1i. To improve gating on these cells, previous work has shown that (i) CD8⁺ CD122⁺ can be much better identified when also including CD44 in the staining mix, which allows gating on CD8⁺ T cells followed by a dot plots of CD44 vs. CD122 where CD44^{hi} CD122^{hi} CD8⁺ T cells can be easily identified. Moreover, such staining and gating allows also the identification of CD44^{hi} CD122^{lo} CD8⁺ T cells, which have been reported to not depend on gc cytokines for their homeostatic proliferation (PMID: 16818671 and 22343569). The authors should provide assessment of these two CD44^{hi} CD8⁺ T cell subsets in their KI mice.

2) Related to my above-mentioned point, it would be very informative and complementary if the authors provided an analysis of Ki67 or BrdU staining of lymphocyte subsets (i.e. CD8⁺ T cell subsets, regulatory T cells and NK cells) directly *ex vivo* in order to provide information on homeostatic, gc cytokine-driven proliferation of these lymphocyte subsets *in vivo*. Such analysis could be complemented by treating animals with native or modified IL-2.

3) In their assessment of NK cells, the authors gated on CD3⁻ lymphocytes and further gated on NK1.1⁺ CD122^{hi} cells. Such gating is valid in WT animals, however, as the authors also showed in this present manuscript, CD122 surface levels are reduced in KI mice, which might affect identification of NK cells by gating on NK1.1⁺ CD122^{hi} cells. The authors should provide a staining that allows identification of NK cells based on double staining with NK1.1, DX5, NKp46 or similar.

4) The summary figure 7 is very informative. Could the authors also integrate their thoughts and previous findings on IL-2R subunit recycling as well as STAT5 dimerization and tetramerization?

Reviewer #3 (IL-2 biology) (Remarks to the Author):

This is a detailed and well-performed study concerning the role of STAT5 phosphorylation in IL-2 driven proliferation of CD8⁺ T cells in vitro. The authors approached this issue by developing novel mouse models where the key tyrosine phosphorylation sites of STAT5a and STAT5b, that lead to STAT5 homo- and heterodimers, were mutated to phenylalanine. The authors note the numbers of T cells and NK cells were diminished, particularly Tregs and memory-phenotypic CD8⁺ T cells. To evaluate the underlying molecular basis by which pSTAT5a and STAT5b controls CD8⁺ T cells, these cells were purified and stimulated with high-dose IL-2 in vitro where global gene expression profiling, STAT5 chromatin binding regions, and the global proteome was determined. The resulting data provide a detailed molecular map related to the role of IL-2-driven proliferation and the relative contribution to STAT5a and STAT5b to this process. This study extends past work where the role of STAT5a and STAT5b were individually knocked out. The study adds to our understanding of how STAT5a and STAT5b phosphorylation and generation of homo- and hetero-dimers are critical for full IL-2 activities in CD8⁺ T cells.

One disappointing aspect of this study is the authors' failure to capitalize on an opportunity to define the role of STAT5 phosphorylation in the homeostasis of T and NK cells. They clearly show that the mutant mice have diminished levels of these cells. Ex vivo studies of these cells in relationship to those from wild-type mice would have been welcomed to establish the physiological contribution of STAT5-dependent gene regulation in maintaining these cells in vivo.

Response to reviewers:

Manuscript NCOMMS-23-59772

Below, we are providing a point-by-point response (in blue) to the reviewers' comments (in black). Similarly, in the manuscript, our changes are now in blue color highlighting.

We have added one author, Dr. Rosanne Spolski, who helped with the new experiment shown in Fig. 1k.

Reviewer #1 (T cell activation)(Remarks to the Author):

The manuscript by Lin et al. addresses the role of STAT5 phosphorylation in vivo. They generated knock-in mice with mutated phosphorylation sites in STAT5A and STAT5B, respectively. They observed reduced numbers of lymphocytes in the homozygous knock-in mice with the strongest phenotype in CD8⁺ T cells (numbers and phenotype of these cells in vivo, response to IL-2 ex vivo). The subsequent omics experiments (RNAseq, proteomics, ChIPseq) revealed substantial defects in the IL-2 signaling on the basis of individual genes. This article reveals the importance of STAT5 phosphorylation in primary cells, documents the dependency of antigen-inexperienced memory-like T cells (aka virtual memory) on STAT5 signaling, shows that STAT5 stimulates other IL-2-induced pathways such as ERK and AKT (probably indirectly), suggests that STAT5B might be more active than STAT5A, and last, but not least, provides very useful resources of IL-2 and/or STAT5 target genes, which will be probably heavily used by the community (including my lab).

The experiments are very well planned and executed. The conclusions are supported by the strong experimental evidence. The data are presented in a very clear way. Overall, I have to congratulate the authors on a very nice and appealing work and I recommend it for publication in Nature Communications.

We are very pleased with the Reviewer's extremely positive comments on our study and that publication in Nature Communications is recommended.

Minor comments:

1. I do not find it necessary to refer to figure details (such as lanes, top/middle panels) in the main text. The figures are self-explanatory and the figure legend might be more appropriate for these detailed explanations, if necessary.

We appreciate the Reviewer's comments, but we prefer to walk the readers through the figures if that is okay. We are simply striving to be as helpful as possible to readers.

2. Line 118 - there is a comma instead of a point at the end of the sentence.

Thanks very much. This has now been fixed.

3. Figure 4a-c - It should be indicated for how long the cells used for these analyses were stimulated with IL-2 (or is it all time points combined?).

We thank the Reviewer for this comment. As requested, we now provide the time of IL-2 stimulation for **Fig. 4a-c** in the legend.

4. The authors claim that "(...) antigen naïve virtual memory cells, which require (...) IL-7 for their maintenance" and cite two papers by Akue et al. and Quinn et al. However, these two papers do not address the importance of IL-7 for virtual memory T cells.

We thank the Reviewer for noting this. We have now modified the text and added appropriate references.

Reviewer #2 (IL-2 biology)(Remarks to the Author):

The authors used CRISPR-Cas9 gene editing to generate Stat5a(Y694F) and Stat5b(Y699F) tyrosine mutant knockin (KI) mice, both in homozygous and heterozygous constellation. When analyzing the animals, they observed similar frequencies of CD4⁺ and CD8⁺ thymocytes in homo- and heterozygous KI mice, but lower CD8⁺ number in homozygous KI mice. Total splenic cell counts were lower in all KI mice, resulting in altered counts of T and B cells compared to wildtype (WT) animals. Frequencies and cell counts of CD4⁺ CD25⁺ T cells, CD8⁺ CD122^{hi} T cells and CD8⁺ CD44^{hi} CD122^{hi} CD49d^{lo} T cells were decreased in all KI mouse lines. Compared to Stat5 knockout (KO) mice, more severe defects of T cell counts were observed in KI mice, suggesting a dominant negative effect of the KI constructs. Moreover, all KI mice showed decreased bone marrow and decreased mature NK cell counts compared to WT animals. When comparing IL-2- and TCR-induced proliferation of CD8⁺ T cells, the authors observed all KI T cells proliferated less upon IL-2 stimulation, whereas TCR-induced proliferation was normal. They performed RNA-seq analysis of WT and KI CD8⁺ T cells at different timepoints after IL-2 stimulation and observed several differentially expressed genes (DEGs) in WT and KI cells; the top ten hallmark genesets dysregulated in both KI mutants were related to cell cycle progression (also confirmed by assessment of cell cycle proteins), and expression of known IL-2-inducible genes were reduced in the KI mutants. Finally, using mass spectrometry-based proteomics, the authors examined the abundance of proteins upon IL-2 stimulation of WT and KI mutant cells, identifying 524 differentially expressed proteins, of which 486 were significantly lower in KI mutant cells compared to WT. These proteomics data were consistent with the RNA-seq data of the authors.

Overall, this is a very well written and nicely presented manuscript summarizing a significant body of work, which provides novel insights into the biology of the STAT5 signaling pathway in vivo in mice. Below are some points to improve the manuscript.

We are very pleased that the Reviewer found our study to be “a very well written and nicely presented manuscript summarizing a significant body of work, which provides novel insights into the biology of the STAT5 signaling pathway in vivo in mice”.

1) When assessing CD8⁺ T cells of KI mice, the authors used CD8 vs. CD122 staining and gated on CD8⁺ CD122^{hi} T cells, which are known to be stimulated by IL-2 and IL-15. However, gating on CD8⁺ CD122^{hi} T cells is not very easy, as also exemplified by the dot plots provided in this manuscript in Fig. 1i. To improve gating on these cells, previous work has shown that (i) CD8⁺ CD122⁺ can be much better identified when also including CD44 in the staining mix, which allows gating on CD8⁺ T cells followed by a dot plots of CD44 vs. CD122 where CD44^{hi} CD122^{hi} CD8⁺ T cells can be easily identified. Moreover, such staining and gating allows also the identification of CD44^{hi} CD122^{lo} CD8⁺ T cells, which have been reported to not depend on gc cytokines for their homeostatic proliferation (PMID: 16818671 and 22343569). The authors should provide assessment of these two CD44^{hi} CD8⁺ T cell subsets in their KI mice.

We thank the Reviewer for this suggestion. We now have added new panels showing CD8⁺CD44^{hi}CD122^{lo} and CD8⁺CD44^{hi}CD122^{hi} cells in WT and *Stat5a* KI and *Stat5b* KI mice (new Fig. 3j and 3k). This is described in a new paragraph on p. 11-12. We also now include the requested references. These new data are consistent with the normal proliferation we see in the mutant CD8⁺ T cells in response to stimulation with anti-CD3 + anti-CD28 but markedly defective proliferation observed in response to IL-2 (Fig. 2a).

2) Related to my above-mentioned point, it would be very informative and complementary if the authors provided an analysis of Ki67 or BrdU staining of lymphocyte subsets (i.e. CD8⁺ T cell subsets, regulatory T cells and NK cells) directly ex vivo in order to provide information on homeostatic, gc cytokine-driven proliferation of these lymphocyte subsets in vivo. Such analysis could be complemented by treating animals with native or modified IL-2.

We thank the Reviewer for this excellent suggestion. We had observed decreased total T cell and CD8⁺ T cell numbers in mutant mice. Consistent with this, when we injected mice with IL-2 and monitored the cell numbers, T cells and CD8⁺ T cells from the mutant mice exhibited poor expansion compared to wild-type mice (new Fig. 1k and 1l, described in a new paragraph on p. 9). We thought it best to directly assess cell numbers as in our experience Ki67 is not a reliable marker. Specifically, in our RNA-seq and mass spectrometry analyses, *Mki67* mRNA (Supplementary Table 3) and Ki67 protein (Supplementary Table 5) are highly induced in WT CD8⁺ T cells but levels were significantly lower in mutant CD8⁺ T cells, so the baseline level is different, and we think the cell count is the best measure of cellular expansion.

3) In their assessment of NK cells, the authors gated on CD3⁻ lymphocytes and further gated on NK1.1⁺ CD122^{hi} cells. Such gating is valid in WT animals, however, as the authors also showed in this present manuscript, CD122 surface levels are reduced in KI mice, which might affect identification of NK cells by gating on NK1.1⁺ CD122^{hi} cells. The authors should provide a staining that allows identification of NK cells based on double staining with NK1.1, DX5, NKp46 or similar.

We thank the Reviewer for this comment/suggestion. We now have performed a new experiment to compare the NK cells by gating CD3⁻NK1.1 combined with CD122, or DX5, or NKp46. Reassuringly, the markers all yielded similar results. We now provide these data in a new Supplementary Fig. 3h and 3i.

4) The summary figure 7 is very informative. Could the authors also integrate their thoughts and previous findings on IL-2R subunit recycling as well as STAT5 dimerization and tetramerization?

We are pleased that the Reviewer likes Figure 7. It is unclear whether IL-2R α subunit recycling would be affected by pY-STAT5 mutants, but it is clear that neither STAT5 dimers nor tetramers would be able to form. We do not think that it will be helpful to add additional information related to this into the Figure. The recycling information may be hard to show and might be confusing, and we hope it is already evident that pTyr is required for dimer formation and thus there also are no tetramers when pTyr is mutated.

Reviewer #3 (IL-2 biology)(Remarks to the Author):

This is a detailed and well-performed study concerning the role of STAT5 phosphorylation in IL-2 driven proliferation of CD8⁺ T cells in vitro. The authors approached this issue by developing novel mouse models where the key tyrosine phosphorylation sites of STAT5a and STAT5b, that lead to STAT5 homo- and heterodimers, were mutated to phenylalanine. The authors note the numbers of T cells and NK cells were diminished, particularly Tregs and memory-phenotypic CD8⁺ T cells. To evaluate the underlying molecular basis by which pSTAT5a and STAT5b controls CD8⁺ T cells, these cells were purified and stimulated with high-dose IL-2 in vitro where global gene expression profiling, STAT5 chromatin binding regions, and the global proteome was determined. The resulting data provide a detailed molecular map related to the role of IL-2-driven proliferation and the relative contribution to STAT5a and STAT5b to this process. This study extends past work where the role of STAT5a and STAT5b were individually knocked out. The study adds to our understanding of how STAT5a and STAT5b phosphorylation and generation of homo- and hetero-dimers are critical for full IL-2 activities in CD8⁺ T cells.

We are very pleased that the Reviewer found our study to be “a detailed and well-performed study... that adds to our understanding of how STAT5a and STAT5b phosphorylation and generation of homo- and hetero-dimers are critical for full IL-2 activities in CD8⁺ T cells”.

One disappointing aspect of this study is the authors' failure to capitalize on an opportunity to define the role of STAT5 phosphorylation in the homeostasis of T and NK cells. They clearly show that the mutant mice have diminished levels of these cells. Ex vivo studies of these cells in relationship to those from wild-type mice would have been welcomed to establish the physiological contribution of STAT5-dependent gene regulation in maintaining these cells in vivo.

We appreciate the Reviewer's comment. We had shown that the pSTAT5 mutant mice have fewer T cells. In accord with Reviewer 2, we now have performed an experiment in which mice were injected with IL-2 followed by ex vivo analysis of T cell numbers (new **Fig. 1k** and **1l**); this is discussed in a new paragraph on p. 9 of the manuscript.

In addition to the requested changes, we now have added three additional supplementary figures (Supplementary Figures 8, 9, and 10) to show flow cytometric gating strategies used in this study.

We thank all the Reviewers for their time and valuable suggestions and believe the revised manuscript is improved. We hope that the manuscript is now acceptable for publication in Nature Communications.

REVIEWERS' COMMENTS

Reviewer #1 (Remarks to the Author):

The authors addressed all my comments. I congratulate the authors and endorse the publication of the manuscript.

Reviewer #2 (Remarks to the Author):

The authors have addressed my previous points.

Reviewer #3 (Remarks to the Author):

No new comments.